

**Locally Produced Sedimentary Biomarkers in High-Altitude Catchments Outweigh**

**Upstream River Transport in Sedimentary Archives**

Alex Brittingham[1,2], Michael T. Hren[3], Samuel Spitzschuch[1], Phil Glauberman[4,5,6], Yonaton

Goldsmith[2], Boris Gasparyan[5] and Ariel Malinsky-Buller[7]

1: Department of Anthropology, University of Connecticut, Storrs, Connecticut, USA

2: The Fredy & Nadine Herrmann Institute of Earth Sciences, The Hebrew University of

Jerusalem, Jerusalem, Israel

3: Department of Earth Sciences, University of Connecticut, Storrs, Connecticut, USA

4: The Catalan Institute of Human Paleoecology and Social Evolution (IPHES) and Universitat

Rovirai I Virgili, Tarragona, Spain

5: Institute of Archaeology and Ethnography, National Academy of Sciences of the Republic of

Armenia, Yerevan, Armenia

6: Department of Early Prehistory and Quaternary Ecology, University of Tübingen, Tübingen,

Germany

7: The Institute of Archaeology, The Hebrew University of Jerusalem, Jerusalem, Israel

*Correspondence to*: Alex Brittingham, alexander.brittingham@mail.huji.ac.il





**Abstract:** Sedimentary records of lipid biomarkers such as leaf wax *n*-alkanes are not only
influenced by ecosystem turnover and physiological changes in plants, they are also influenced by
earth surface processes integrating these signals. The integration of biomarkers into the
sedimentary record and the effects of integration processes on recorded environmental signals are
complex and not fully understood. To determine the depositional constraints on biomarker records
in a high-altitude small catchment system, we collected both soil and stream sediments along a
1000 m altitude transect (1500 – 2500 masl) in the Areguni Mountains, a subrange of the Lesser
Caucasus Mountains in Armenia. We utilize the existence of a treeline at ~ 2000 masl, which
separates alpine meadow above from deciduous forest below, to assess the relative contribution of
upstream biomarker transport to local vegetation input in the stream. We find that average chain
length (ACL), hydrogen isotope ($\delta$D) and carbon isotope ($\delta^{13}$C) values of *n*-alkanes are
significantly different in soils collected above and below the treeline. However, samples collected
from the stream sediments do not integrate these signals quantitively. As the stream drops below
the treeline, the ACL, $\delta$D and $\delta^{13}$C values of *n*-alkanes preserved in streambed sediments reflect a
bias toward *n*-alkanes sourced from trees. This suggests that there is either 1) minimal
transportation of organic matter from the more open vegetation in higher elevations, or 2) greater
production of target biomarkers by trees and shurbs found at lower elevations results in
overprinting of stream signals by local vegetation. Though this latter observation may preclude
using *n*-alkanes to measure past treeline movement in these mountains, $\delta$D values of biomarkers
in fluvial deposits in these settings are more likely to record local hydrological changes rather than
changes in upstream fractionation differences associated with vegetation turnover.



## 1. Introduction

Mountain regions are important hubs for biodiversity and can provide refuge for a number of endemic species of flora and fauna (Antonelli et al., 2018). However, these high-altitude environments are often particularly vulnerable to climate change (Guisan and Theurillat, 2000). Therefore, gaining an understanding of sensitivity of these regions to past climate change is important for projecting the effects of future climate change on fragile ecosystems. The so called Caucasus Region in particular has been identified as a biodiversity hotspot covering the Republics of Armenia, Georgia, Azerbaijan, and parts of the Russian Federation, Türkiye, and Iran, supports a wide variety of plant and animal species (Zazanashvili, 2009; Gasparyan and Glauberman, 2022). To better understand climate and environmental change in both the past and the present, it is necessary to refine our understanding and interpretation of paleoclimate records in this region. Specifically, we are interested in understanding the sedimentary processes involved in the formation, transport, recycling, and accumulation of organic biomarkers in sedimentary archives and assessing whether these archives record a local environmental signal or are a mix of local and transported organic material.

Normal alkanes ($n$-alkanes) are an important component of the epicuticular wax in terrestrial plants. This waxy coating on plants protects against ultraviolet damage, water loss and predation (Jetter et al., 2006). Specific compounds in this wax, such as $n$-alkanes, are a useful tool for reconstructing past environmental changes through the analysis of the distribution of alkane homologues as well as their stable hydrogen ($\delta D$) and carbon ($\delta^{13}C$) isotope values. Previous research in the Greater and Lesser Caucasus Mountains has documented the applicability of the average chain length (ACL) of leaf wax biomarkers as a tool for differentiating between grassy



and deciduous vegetation (Bliedtner et al., 2018; Trigui et al., 2019), though on a global scale ACL
does not differentiate well between vegetation types (Bush and McInerney, 2013).

The biggest driver of the carbon isotope ($\delta^{13}$C) values of plant tissue is the photosynthetic

pathway of the plant (Diefendorf and Freimuth, 2017). $C_3$ plants, which thrive in areas with cooler
growing season temperatures, have more negative $\delta^{13}$C values than do $C_4$ plants, which thrive in
warmer growing season temperatures (Ehleringer et al., 1977). $C_3$ vegetation is further influenced
by water use efficiency, as water stress influences the $c_i/c_a$ ratio of plants (Farquhar et al., 1982).
$\delta^{13}$C values in lipids generally follow the same trends, and $C_3$ plants have more negative $\delta^{13}$C lipid
values than $C_4$ plants (Diefendorf and Freimuth, 2017). However, carbon fractionation of lipids is
not consistent in different classes of plants (Pedentchouk et al., 2008; Sikes et al., 2013; Diefendorf
et al., 2011).

The hydrogen isotope ($\delta$D) values of *n*-alkanes in terrestrial plants record the $\delta$D values of

environmental water (Sachse et al., 2012). This is typically reflective of $\delta$D values in precipitation,
though precipitation $\delta$D values can also undergo positive shifts due to soil evaporation. The $\delta$D
values of plant waxes are also influenced by fractionation during biological synthesis of lipids,
which imparts a strong negative fractionation on $\delta$D values, as well as transpiration of leaf water
(Gamarra et al., 2016).The fractionation between meteoric water and lipids is typically larger in
gymnosperms than in angiosperms (Pedentchouk et al., 2008; Oakes and Hren, 2016).

Despite the benefits in measuring $\delta$D and $\delta^{13}$C values in *n*-alkanes for understanding

environmental and hydrological processes, not all the processes modifying isotope values from
plant to *n*-alkane deposition are well understood. Sedimentary integration is one of the most poorly
understood aspects of this process (Sachse et al., 2012). A number of studies on the integration of
leaf waxes in catchments have been published in recent years which help clarify these processes



(Alewell et al., 2016; Feakins et al., 2018a; Häggi et al., 2016a; Hemingway et al., 2016; Ponton
et al., 2014; Suh et al., 2019). However, most of these studies have focused on large river systems
rather than first order streams. Thus, the sedimentary processes involved in the formation,
transport, recycling, and accumulation of organic biomarkers in first and second order streams are
not well understood. One challenge in assessing these processes in small streams is that the
environment and plant communities are often homogenous, and thus it is not possible to
differentiate between local and upstream transported organic material. To better understand
transport processes affecting organic material in small catchments, we studied a set of streams in
the Dany River, a tributary of the Barepat River, located in the Areguni Mountains in the Lesser
Caucasus Range. This stream system is divided into two distinct ecological regions by the treeline
(at ~ 2000 masl), which separates alpine meadow above the tree line (2000 – 2500 masl) from
deciduous forest below (1500 – 2000 masl). To evaluate the input of *n*-alkanes from upstream
transported organic material relative to vegetation near the stream, we collected soil samples on
the slopes of the mountains from both above- and below the treeline throughout the watershed and
sediments deposited in the streambed along an elevation transect. Comparing the hillslope
sedimentary biomarkers and the streambed sedimentary biomarkers allows assessment of the input
of *n*-alkanes locally produced by vegetation compared to those transported in stream sediments
within the catchment.

An additional motivation of this research is that treelines are a vulnerable feature of higher

altitude environments. Previous research in the Areguni Mountains study area has assessed the
relationship between treeline dynamics and climate forcing in the past (Ghukasyan et al., 2010;
Montoya et al., 2013; Malinsky-Buller et al., 2021; Tornero et al., 2016), and the Pleistocene
sediments uncovered at archaeological sites at Kalavan village within this area have the potential



to reconstruct this relationship. However, in order to reconstruct these systems in the past it is
important to understand modern biomarkers integration processes in the first and second order
streams and their potential effects on the sedimentary archives of the Areguni Mountains.
**2. Methods**
**2.1 Sample Collection and Extraction**
Hillslope soil samples were collected in September 2018 along an altitude transect (1500 – 2500
masl) above the Dany River watershed, a first order tributary of the Barepat River in the Areguni
Mountains, Armenia (Fig 1), which traverses the treeline at ~2000 masl. Soil samples were
collected by first clearing the top ~10 cm of soil to remove roots. Stream bed sediment samples
were collected from the Dany River throughout the altitude transect at intervals of ~100 m in
altitude. In all cases, roughly 100 g of sediment were collected for extraction of *n*-alkanes. In order
to extract *n*-alkanes, samples were extracted using a Soxhlet apparatus with 2:1
dichloromethane:methanol for 48 hours. Following lipid extraction, *n*-alkanes were separated with
silica gel column chromatography and quantified on a Thermo-Scientific Trace GC Ultra
(Manufacturer) fitted with a split–splitless (SSL) injector and flame ionization detector (FID) using
a BP-5 column (30 m × 0.25 mm i.d., 0.25 μm film thickness) with He as the carrier (1.5 ml/min).
Odd over even predominance (OEP) (Eq. 1) and average chain length (ACL) (Eq. 2) were used to
evaluate distributions of *n*-alkanes (REF). We also calculated $P_{aq}$, an *n*-alkane proxy to evaluate
the possible biomarker contribution of aquatic and emergent plants (Eq. 3) (Ficken et al., 2000).
$$OEP = \frac{C_{25} + C_{27} + C_{29} + C_{31} + C_{33}}{C_{24} + C_{26} + C_{28} + C_{30} + C_{32}}$$
$$ACL = \frac{25 * C_{25} + 27 * C_{27} + 29 * C_{29} + 31 * C_{31} + 33 * C_{33}}{C_{25} + C_{27} + C_{29} + C_{31} + C_{33}}$$



$$Paq = \frac{C_{23} + C_{25}}{C_{23} + C_{25} + C_{29} + C_{31}}$$
**2.3 Stable Isotope Analysis**
δD and $\delta^{13}C$ values of individual *n*-alkanes were measured with a Thermo GC-Isolink coupled
with a Thermo Scientific MAT 253 (manufacturer) isotope ratio mass spectrometer with a BP-5
column (30 m × 0.25 mm i.d., 0.25 μm film thickness). Oven temperature was set at 50°C for 1
min, ramped to 180°C at 12°C/min, then ramped to 320°C at 6°C/min and held for 4 min. Internal
standards (Mix A5 from A. Schimmelman) were run every four samples across a range of
concentrations to correct for size effects. Standard deviations were 0.5‰ for $\delta^{13}C$ and 4‰ for δD.
Isotope ratios (R) were converted to δX ($\delta^{13}C$ and δD) values (Eq. 3) and are expressed in permill
(‰).
$$\delta X = \left( \frac{R_{Sample}}{R_{Standard}} - 1 \right) * 1000$$

**3. Results**
**3.1. Alkane abundances**
The most abundant alkane homolog in samples collected in the Areguni Mountains is the
$C_{29}$ or $C_{31}$ alkane, which is typical for terrestrial plants. Odd numbered alkanes are significantly
more abundant than even numbered alkanes, and the OEP of all samples averages 11.2, with a
range from 7.4-18.4. There is no significant difference between the mean OEP of soil (11.1) and
stream (11.3) samples in the watershed. These values are similar to those previously measured in
the Greater and Lesser Caucasus Mountains (Trigui et al., 2019; Bliedtner et al., 2018).



The mean average chain length (ACL) of all samples averages 29.7, with a range from 28.4
to 31.8 (Fig 3). In soils above the treeline, the average ACL value is 30.6 (range of 29.8-31.8). In
soils below the treeline, the average ACL value is 29.5 (range of 28.4-30.4). There is a significant
difference between the average ACL values of the above treeline and below treeline soils
($p<0.001$). Stream sediment above the treeline has an average ACL value of 29.7 (range of 29.1-
30.2) and stream sediments below the treeline have an average ACL value of 29.3 (range of 28.6
to 30.0). The stream sediments from below the treeline have a significantly ($p<0.001$) lower
average ACL value than those above the treeline.
The $P_{aq}$ values of *n*-alkanes in these samples suggests a mostly terrestrial origin of the
organic matter. Higher $P_{aq}$ values indicate contributions of floating and emergent macrophytes.
However, we do not find a significant difference between the $P_{aq}$ values in the stream sediments
when compared to the soil samples, indicating that the organic load of the stream sediments is
mostly of terrestrial origin. Terrestrial plants have average $P_{aq}$ values of 0.09, with emergent plants
averaging 0.25 (Ficken et al., 2000). Only eight of the 51 samples in this study had $P_{aq}$ values above
0.20, four stream and four soil samples. This indicates that there was not a significant contribution
of aquatic plants in the Dany stream sediments, and the biomarker load is primarily terrestrial in
origin.

**3.2. δD and δ$^{13}$C values**

The δ$^{13}$C values in soils and stream sediments collected from the Areguni Mountains reflect
a $C_3$ landscape, which is typical in Armenia. δ$^{13}$C values in all samples ranged from -36.0 to -
32.3‰ (Fig 4). The range is similar for both soil samples (-35.9 to -32.3‰) and stream samples (-
36.0 to –32.5‰). However, there is a significant difference in the δ$^{13}$C values of above and below
treeline samples, both in the stream and soil samples collected. Above the treeline, δ$^{13}$C values in



soils averages -34.9‰, and below the treeline soil alkanes average -33.3‰. Stream sediment $\delta^{13}$C
values average -35.0‰ above the treeline and -33.6‰ below the treeline. In both cases, these
values are significant (p<0.0001, student's t-test). $\Delta^{13}$C values in stream samples exhibit a step-
like behavior, with ~2‰ shift to more negative values as the stream drops below the treeline.

The $\delta$D values measured in soil samples collected in the catchment ranged from -144 to -

185‰ (Fig 5). These values were significantly (p<0.001, student's t-test). More negative in above
treeline sediments (-175‰) than in below treeline sediments (-156‰). This is also true in sediment
collected from stream samples, which are significantly more negative above the treeline (-175‰)
than below the treeline (-158‰).  As with the $\delta^{13}$C values, the $\delta$D values of stream samples show
sudden change as the stream drops below the treeline.
**4. Discussion**
**4.1 Integration of local and upstream soil *n*-alkanes into the river sediments**

The hillslope soil leaf wax $\delta$D, $\delta^{13}$C and ACL show a step-like change at the treeline,

indicating a significant separation between upstream (above treeline) and downstream (below
treeline) soils. Using this separation, it is possible to assess the contributions and integration of
upstream vs. downstream soils to the streambed sediments along the altitude transect. The step-
like transition in streambed $\delta$D and $\delta^{13}$C values indicates an over-printing of upstream alkane
isotope values by input from deciduous vegetation. Thus, local production largely outweighs
upstream transport in this setting. However, to firmly evaluate the upstream and downstream
hillslope soil contribution to streambed sediments, there is a need to quantitatively evaluate the
area-weighted production of *n*-alkanes above and below the treeline.
**4.2. Modeling *n*-alkane production and estimating upstream transport and integration**



To further evaluate the integration of *n*-alkanes above and below the treeline, we created a
mixing model that calculates the expected δD, δ$^{13}$C and ACL values at each one of the sampling
locations based on the *n*-alkane production of hillslope sediments above each streambed sampling
point (Fig. 6). This mixing model assumes that the *n*-alkanes in the river are a function of the
weighted *n*-alkane production above the sampling location.
The parameters we used for this mixing model are: 1. Satellite images to map the areas of
tree and grass sediment throughout the Dany River catchment. 2. An estimate of net primary
productivity of organic material production in grasses and trees (grams per area) (Brun et al.,
2022). 3. Estimates of *n*-alkane production in grasses and trees in the Greater and Lesser Caucasus
Mountains (grams of *n*-alkane per gram of organic material) (Trigui et al., 2019; Bliedtner et al.,
2018). 4. End member values of δD, δ$^{13}$C and ACL derived from the average hillslope soils above
and below the treeline. By multiplying these terms (area x organic mass production x *n*-alkane
production x end member soils value), we created an *n*-alkane production map for the Dany River
catchment. Using this map, we calculated, for each riverbed sampling location, the amount of grass
and tree *n*-alkanes produced on the hillslopes above the sampling locations.
We compared the results of this mixing model with the measured δD, δ$^{13}$C and ACL in the
streams. Stream sediment samples collected above the treeline (from ~2000-2600 masl) fall within
the range of expected values, however, samples below the treeline consistently over-sample
deciduous-sourced *n*-alkanes. Measured values do not have a linear relationship with the expected
values based on vegetation area. These measured values would produce under-estimates of the
upstream area of alpine grasses, yielding incorrect reconstructions of paleo-vegetation in
sedimentary records. Comparing the mixing model with the observations indicates that an area-
weighted mixing process is not an adequate model for explaining the *n*-alkanes signal in the





streambed sediments. A simple and straightforward way to interpret this discrepancy is that an
area-weighted quantitative integration of *n*-alkanes is not a good model for describing this
catchment system, and that local production is much larger than transported organic material.

However, there are still other factors that may be driving this process that our mixing model

does not account for. First, the average slope of forested areas in the Dany watershed is higher than
those in grassy areas. These steeper slopes would cause more sediment transport into the stream
bed. Second, though production of *n*-alkanes in grasses and trees is not significantly different in
the Greater and Lesser Caucasus Mountains, concentrations are higher in soils in deciduous areas
(Trigui et al., 2019; Bliedtner et al., 2018). This retention of more biomarkers in forest soils would
also increase the contribution of deciduous alkanes into the stream bed. Third, stream downcutting
into older sediments has the potential to re-mobilize stored organic carbon, which may contain a
greater load of deciduous *n*-alkanes. However, analysis of pollen from a nearby lake core  in the
Areguni Mountains shows a gradual shift over the last 4000 years from a grass-dominated
landscape to the deciduous forest present today (Joannin et al., 2022). Therefore, stored biomarkers
are more likely to be grass-dominant, and this is unlikely to explain the measured bias to deciduous
alkanes.

Since *n*-alkanes in this first order stream do not quantitively integrate *n*-alkanes based on

the upstream area of different vegetation types, this likely precludes the use of *n*-alkanes as a tool
to reconstruct vertical treeline movement in this setting. However, this is a benefit for attempts to
reconstruct hydrological changes through the analysis of δD values in *n*-alkanes. Given the ~20‰
difference in apparent fractionation (ε) values for above and below treeline sediments, changes in
upstream vegetation cover would alter measured δD values in *n*-alkanes in sedimentary archives.
Without this quantitative integration, *n*-alkanes measured in the Pleistocene sediments found in





this watershed are more likely to reflect changes in δD values of precipitation, and therefore would
serve to reconstruct hydrological cycles, rather than changes in upstream vegetation cover. Since
$\delta^{13}C$ and ACL of *n*-alkanes are also different in above and below treeline sediments, these other
analyses would also be useful to identify periods with large changes in treeline that might
complicate interpretation of δD values.
In order to illustrate this point, we present hypothetical records of biomarker δD values
from three points in the Dany watershed (Fig. 7) documenting 20‰ and 30‰ shifts in precipitation
δD values. Given the lack of quantitative integration in the catchment, a paleoclimate record from
either above (A) or below (C) treeline would record the shift in precipitation δD values. Below
treeline sedimentary records, with stream organic biomarker load overprinted by local vegetation
production, would likely provide a means to reconstruct the δD precipitation values. However,
records near the treeline (B) would be heavily affected by changes in apparent fractionation values
associated with changes in vegetation around the stream. Co-occurring climate forcing of shifts in
δD values of precipitation and changes in treeline altitude would cause paleoclimate records in this
zone to over-estimate the magnitude of precipitation δD value shifts.
Previous studies on the integration of organic biomarkers has produced mix results, with
some demonstrating spatial integration of catchment signals (Alewell et al., 2016; Hemingway et
al., 2016; Feakins et al., 2018b), whereas others did not observe this (Häggi et al., 2016b; Ponton
et al., 2014). However, these previous studies typically focused on very large river systems, which
will undergo different transport processes than the first-order streams analyzed in this study. A
number of these studies (Alewell et al., 2016; Hemingway et al., 2016; Ponton et al., 2014; Feakins
et al., 2018b) also observed seasonal differences in biomarker load in river sediments. Collecting



seasonal samples in the Areguni Mountains, as well as testing these processes in other first-order
streams, could further help clarify the transport processes measured in this setting.


**5. Conclusion**

Sediment and stream samples from the Areguni Mountains, a subrange of the Lesser Caucasus
Mountains in Armenia, demonstrate that there is a significant difference in hillslope soil δD, $\delta^{13}$C
and ACL values above and below treeline. *n*-alkanes in sediments in the Areguni Mountains can
be used to differentiate between the above and below treeline sediments. However, *n*-alkanes
extracted from stream sediments reflect their local area, rather than demonstrating transport from
the higher-altitude alpine meadow. These results provide a complication for attempts to reconstruct
changes in past treeline in this mountain range, given that the biomarker load in stream does not
reflect the relative area of different upstream vegetation types. However, these results simplify
interpretation of past *n*-alkane δD values, as apparent fractionation differences between grasses
and trees are less likely to impart a significant influence on δD values in stream bed *n*-alkanes.
6. Competing interests
The contact author has declared that none of the authors has any competing interests
7. Acknowledgements
We would like to thank the Kalavan villagers for their help, support, and hospitality: especially
the Ghukasyan family for providing us a home away from home. We also thank Suren Kesejyan,
Hovhannes Partevyan, and Vardan Stepanyan. The research in Kalavan project was funded by the
support of The Gerda Henkel Stiftung grant n. AZ 10_V_17 and n. AZ 23/F/19, the Leakey
Foundation. AB is thankful to the Lady Davis foundation, Fritz-Thyssen Foundation grant awarded



for the project "Pleistocene Hunter-Gatherer Lifeways and Population Dynamics in the Ararat
(paleo-lake) Depression, Armenia", and The European Research Council grant N 948015:
"Investigating Pleistocene population dynamics in the Southern Caucasus" (awarded to AMB) for
current financial support. Further support was provided by "Areni-1 Cave" Consortium ["Areni-1
Cave" Scientific-Research Foundation (Armenia), and the "Gfoeller Renaissance Foundation"
(USA)], as well as the Institute of Archaeology and Ethnography of the National Academy of
Sciences of the Republic of Armenia (supported by the Higher Education and Science Committee,
Republic of Armenia, under grant number 21AG-6A080).



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



Figure Captions
Figure 1: (Left) Topographic map of Armenia with inset map of sampling location (black box)
(Right) Inset map of soil (yellow circles) and stream (blue circles) samples collected in the
Areguni Mountains, along with the limit of the Barepat (dashed line) and Dany watersheds (solid
line)
Figure 2: The δD and δ$^{13}$C values of *n*-alkanes extracted from above treeline (green squares) and
below treeline (red triangles) sediments
Figure 3: The average chain length (ACL) values of *n*-alkanes extracted from above treeline
(green squares) and below treeline (red squares) and stream (blue triangles) sediments across the
sampling elevation gradient
Figure 4: The δ$^{13}$C values of *n*-alkanes extracted from above treeline (green squares) and below
treeline (red squares) and stream (blue triangles) sediments across the sampling elevation
gradient
Figure 5: The δD values of *n*-alkanes extracted from above treeline (green squares) and below
treeline (red squares) and stream (blue triangles) sediments across the sampling elevation
gradient
Figure 6: Comparison of the measured ACL, δD and δ$^{13}$C values against expected values of
stream sediments. Dashed line represents the range of expected values from stream sediments if
vegetation was integrated equally by area
Figure 7:  A photograph of the Dany watershed with hypothetical paleoclimate record from three
locations: (A, dashed line) Below treeline, (B, solid line) near treeline with fluctuations in
treeline altitude, and (C, dotted line) above treeline








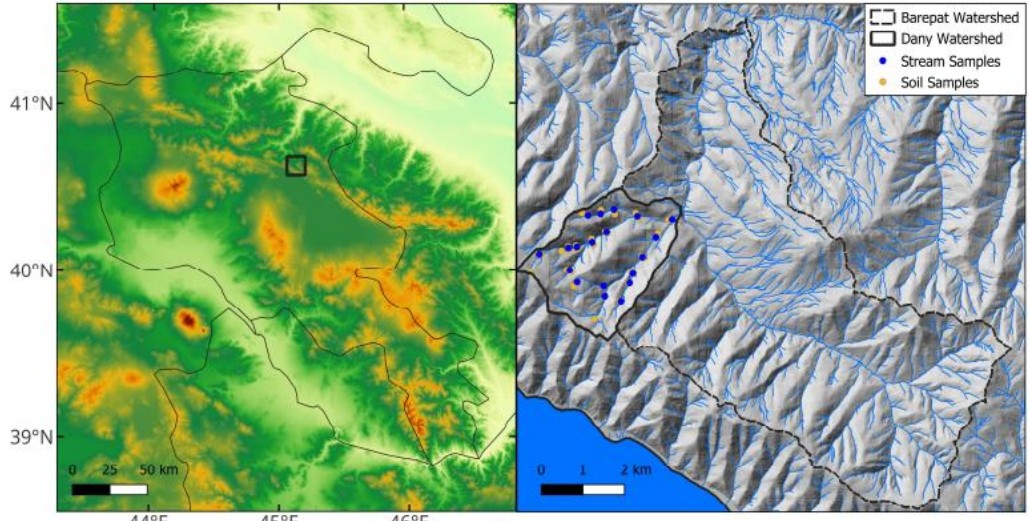

Figure 1: (Left) Topographic map of Armenia with inset map of sampling location (black box) (Right) Inset map of soil (yellow circles) and stream (blue circles) samples collected in the Areguni Mountains, along with the limit of the Barepat (dashed line) and Dany watersheds (solid line)






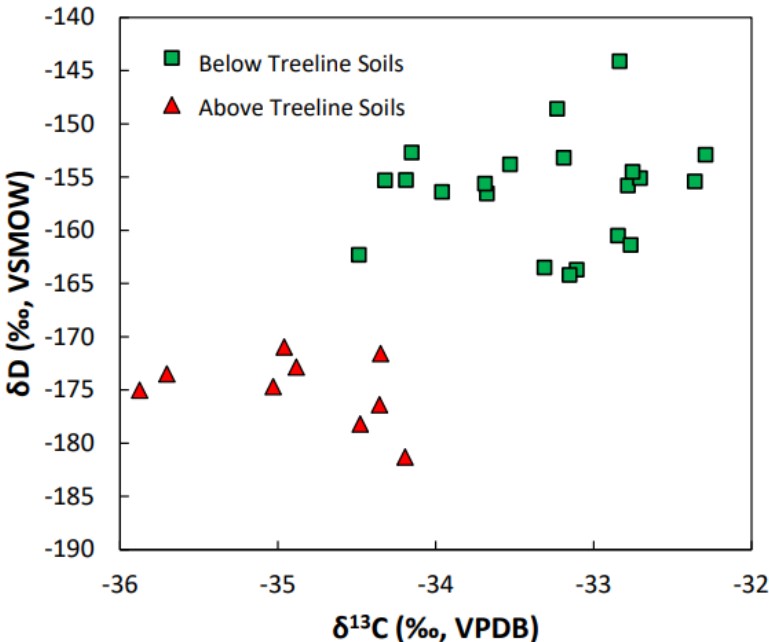

Figure 2: The δD and δ¹³C values of *n*-alkanes extracted from above treeline (green squares) and below treeline (red triangles) sediments





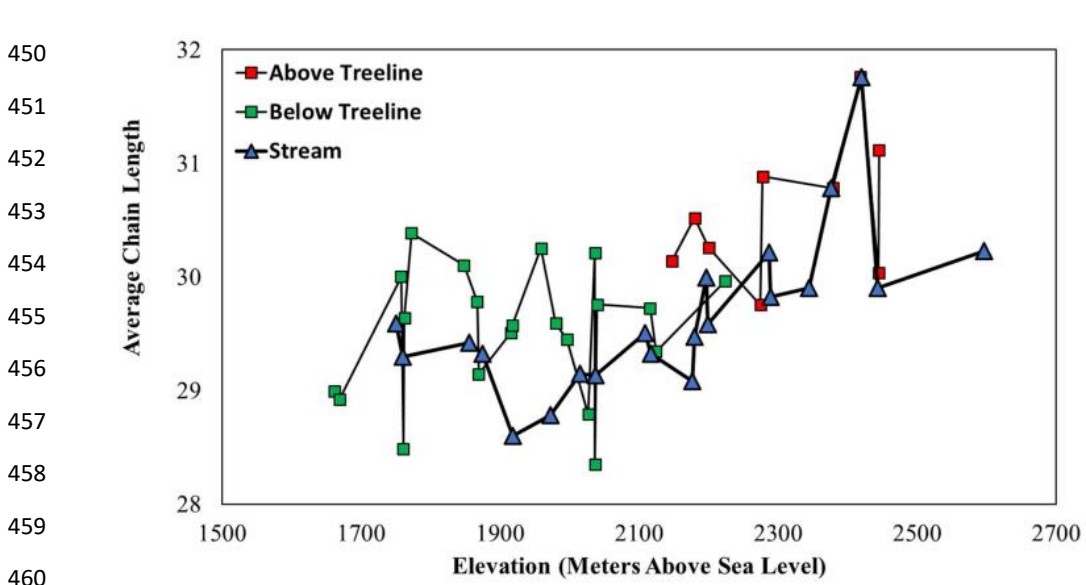

Figure 3: The average chain length (ACL) values of *n*-alkanes extracted from above treeline (green squares) and below treeline (red squares) and stream (blue triangles) sediments across the sampling elevation gradient







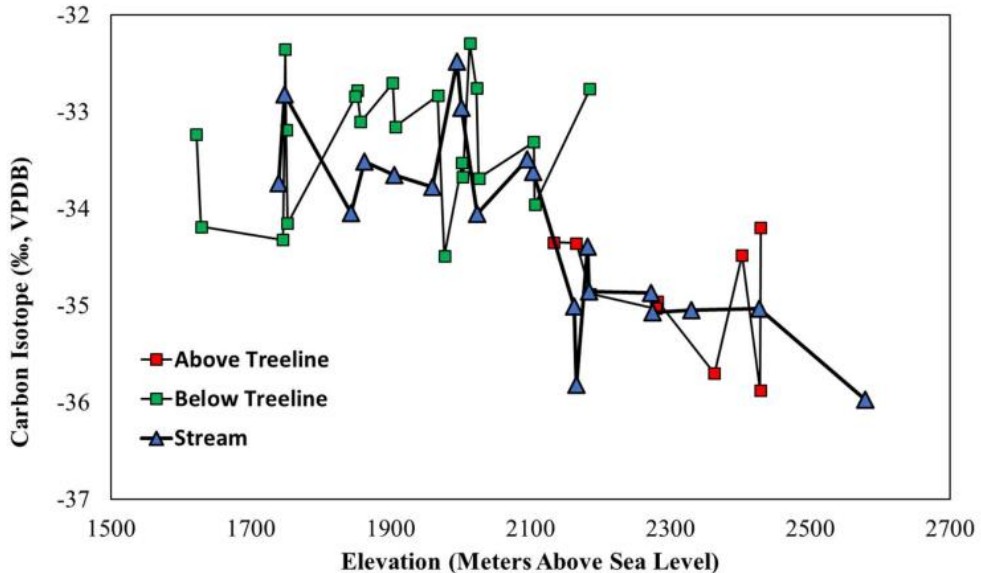

Figure 4: The δ$^{13}$C values of *n*-alkanes extracted from above treeline (green squares) and below treeline (red squares) and stream (blue triangles) sediments across the sampling elevation gradient






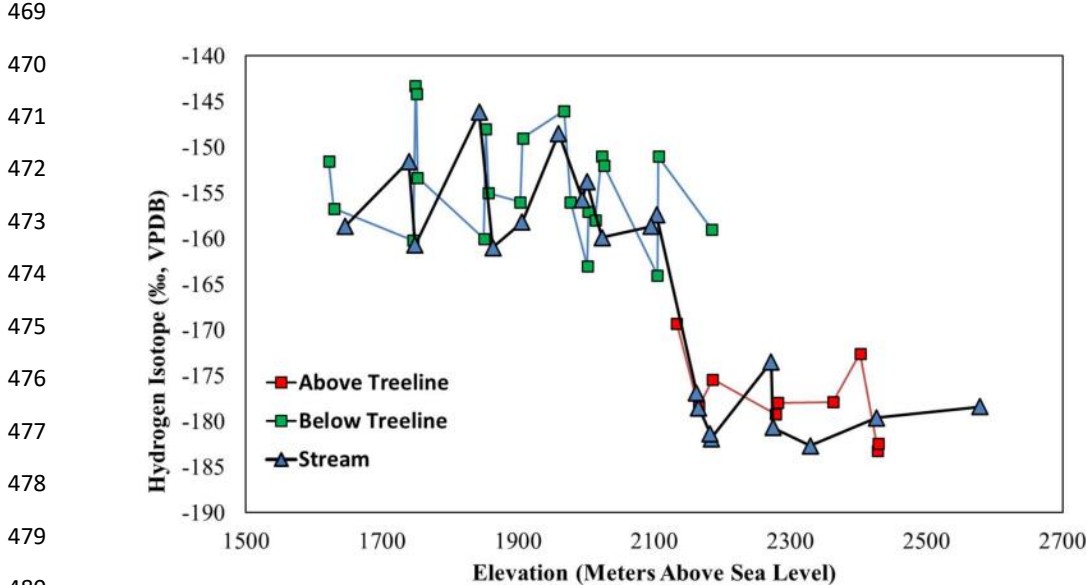

Figure 5: The δD values of *n*-alkanes extracted from above treeline (green squares) and below treeline (red squares) and stream (blue triangles) sediments across the sampling elevation gradient



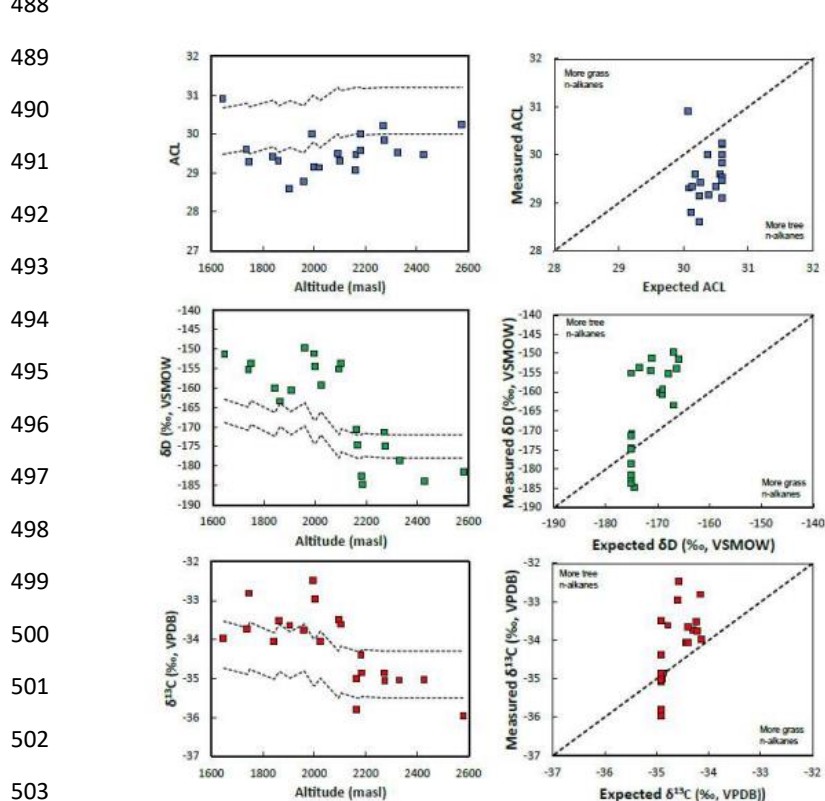

Figure 6: Comparison of the measured ACL, δD and δ¹³C values against expected values of stream sediments. Dashed line represents the range of expected values from stream sediments if vegetation was integrated equally by area



510

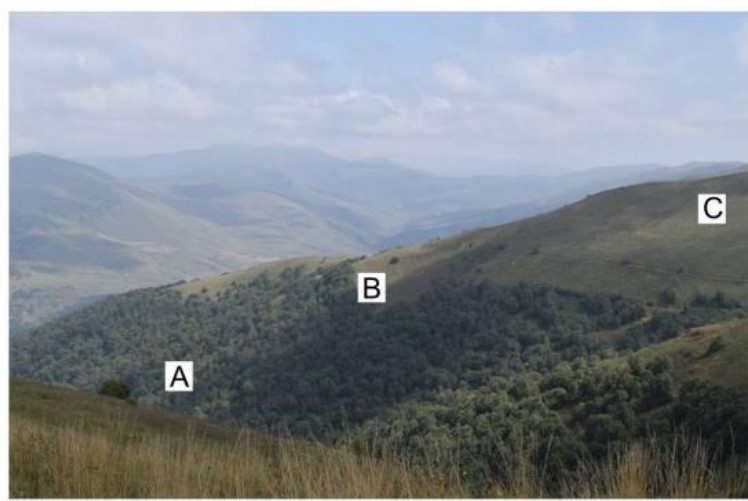

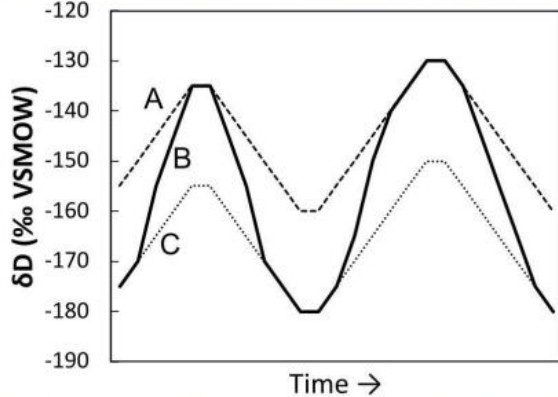

Figure 7: A photograph of the Dany watershed with hypothetical paleoclimate record
from three locations: (A, dashed line) Below treeline, (B, solid line) near treeline with
fluctuations in treeline altitude, and (C, dotted line) above treeline