# Peer review of "Locally Produced Leaf Wax Biomarkers in the High-Altitude Areguni Mountains"

_EGUsphere, 2024_

## Author Comment (AC1)

**Reviewer 1**

**General comments:**

*My main comment on the manuscript is a request that the authors include information about their mixing model, including necessary equations and values used for constants, as a subsection in the methods. The values for constants could be added as a table or in the text. The additional text and equations would substantially clarify the discussion in section **4.2.** and, more importantly, are necessary for evaluating the comparison between measured and expected values shown in Figure 6. Much of the necessary text can be moved from lines 194–208 to the methods section.*

- We agree with the reviewer, an additional section will be added describing the model in detail. We will include a figure detailing the boundaries of the different vegetation types in the catchment. This section will also include a table describing the endmembers used and the expected isotope values at each point in the catchment.

*I also strongly recommend appending the supporting data for this study as supplementary tables or as files uploaded to a publicly accessible database. The data may prove useful for future paleoclimate or palaeoecological work in this region, so it is important to make sure the data is accessible to those who may need it in the future.*

- A supplementary table will be added to the final draft, including isotope values, n-alkane distributions and sampling locations.

*Lastly, I recommend inserting more references to the figures within the text. As is, each figure is only referenced once or twice, even in sections where the data is discussed in detail. More figure references will make the text easier to follow.*

- We will refer to the figures more in the final draft

**Specific comments**

*Title*

*I suggest amending the title to better reflect the findings and conclusions of the study. As is, the title appears to suggest that the findings here are applicable to all high-altitude river (and lake?) catchments. However, this assertion is not made in the manuscript.*

*Two other things to point: (1) n-alkanes measured in this study should really be referred to as leaf wax or vegetation biomarkers since they are produced by plants, not produced by sediments; (2) I believe you mean downstream river transport, as in away from the river's headwaters. As written, the word upstream implies transport of sediment in the opposite direction of the water flow, which I do not think is what you mean.*

*A potential alternative title could be:*

*"Locally Produced Leaf Wax Biomarkers in the High-Altitude Lesser Caucuses Outweigh Downstream Transport"*

- We will alter the title to be more general, and change it as follows: "Locally Produced Leaf Wax Biomarkers in the High-Altitude Areguni Mountains Outweigh Downstream Transport"

**Abstract**

*Lines 18–22: these first two sentences are a bit repetitive. I suggest removing the first sentence that reads, "Sedimentary records…signals." and starting the abstract with the second sentence that reads "The integration of…". This still introduces the study effectively while making the language more succinct.*

- We will revise the abstract to combine these sentetences: "Sedimentary records of lipid biomarkers such as leaf wax n-alkanes are not only influenced by ecosystem turnover and physiological changes in plants, they are also influenced by earth surface processes integrating these signals into the sedimentary record, though the effect of these integration processes are not fully understood."

*Line 25: I suggest removing "the existence of" and just stating "We utilize a treeline…"*

- We will make this change

*Lines 35–36: Please clarify the language here, I am unsure what you mean by the "latter observation" since the language in lines 32–35 makes it seem like you do not have sufficient evidence to distinguish which mechanism is driving the pattern you observe in the stream sediments (this is also what I took away from reading the rest of the manuscript).*

- We will remove the 'latter observation'; since either (1) or (2) would prevent n-alkanes from quantitatively reflecting tree line movement. This sentence will read "Though both of these observations…"

*Line 38: I am unclear what "changes in upstream fractionation differences" means here. I suspect you are talking about fractionation of leaf wax hydrogen isotope composition caused by changes in landscape vegetation, please clarify the language as this confused me a bit.*

- The reviewer is correct that this is in reference to fractionation of leaf wax hydrogen isotope values, to clarify we will change the sentence to read "$\delta D$ values of biomarkers in fluvial deposits in these settings are more likely to record local hydrological changes rather than reflect fractionation changes due to turnover in upstream vegetation structure".

**Introduction**

*Line 45: Is "so called" necessary here?*

- We will remove this

*Line 49: I suggest adding a sentence here briefly summarizing any paleoclimate records from this region that utilize leaf wax n-alkanes if there are any as that will clarify the importance of this study to your readers.*

- We will add references to previous studies in Armenia that have used leaf wax n-alkanes. To the best of our knowledge, the following papers are those that use plant waxes in the region:
  - Brittingham, Alex, Michael T. Hren, Gideon Hartman, Keith N. Wilkinson, Carolina Mallol, Boris Gasparyan, and Daniel S. Adler. "Geochemical evidence for the control of fire by Middle Palaeolithic hominins." *Scientific Reports* 9, no. 1 (2019): 15368.
  - Malinsky-Buller, Ariel, Philip Glauberman, Vincent Ollivier, Tobias Lauer, Rhys Timms, Ellery Frahm, Alexander Brittingham et al. "Short-term occupations at high elevation during the Middle Paleolithic at Kalavan 2 (Republic of Armenia)." *PLoS One* 16, no. 2 (2021): e0245700.
  - Glauberman, Phil, Boris Gasparyan, Jennifer Sherriff, Keith Wilkinson, Bo Li, Monika Knul, Alex Brittingham et al. "Barozh 12: Formation processes of a late Middle Paleolithic open-air site in western Armenia." *Quaternary Science Reviews* 236 (2020): 106276.
  - Malinsky-Buller, Ariel, Lotan Edeltin, Vincent Ollivier, Sébastien Joannin, Odile Peyron, Tobias Lauer, Ellery Frahm et al. "The environmental and cultural background for the reoccupation of the Armenian Highlands after the Last Glacial Maximum: The contribution of Kalavan 6." *Journal of Archaeological Science: Reports* 56 (2024): 104540.
  - Trigui, Yesmine, Daniel Wolf, Lilit Sahakyan, Hayk Hovakimyan, Kristina Sahakyan, Roland Zech, Markus Fuchs, Tilmann Wolpert, Michael Zech, and Dominik Faust. "First calibration and application of leaf wax n-alkane biomarkers in loess-paleosol sequences and modern plants and soils in Armenia." *Geosciences* 9, no. 6 (2019): 263.

*Line 62: I suggest replacing "though" with "although"*

- We will make this change

*Line 64: I suggest rewriting as "The carbon isotope ($\delta^{13}C$) composition of plant tissue is primarily set by the photosynthetic pathway of the plant.*

- We will make this change

*Line 72: Could you please comment on which of these factors is important for your study site? For example, are there $C_4$ plants in the Caucuses?*

- We will add the following sentence in order to describe the importance of C4 vegetation in the study: "Currently, C4 vegetation makes up around 3% of identified species in Armenia (Rudov et al 2020), and was present in the Kalavan region during the Holocene (Tornero et al 2016). "
    - Rudov, Alexander, Marjan Mashkour, Morteza Djamali, and Hossein Akhani. "A review of C4 plants in southwest Asia: an ecological, geographical and taxonomical analysis of a region with high diversity of C4 eudicots." Frontiers in plant science 11 (2020): 546518.
    - Tornero, Carlos, Marie Balasse, Adrian Bălăşescu, Christine Chataigner, Boris Gasparyan, and Cyril Montoya. "The altitudinal mobility of wild sheep at the Epigravettian site of Kalavan 1 (Lesser Caucasus, Armenia): Evidence from a sequential isotopic analysis in tooth enamel." Journal of Human Evolution 97 (2016): 27-36.

*Lines 83–86: I suggest rewriting this sentence to summarize the common findings of the studies you cite here. For example, I recall that the Feakins et al. 2018a study cited here found that riverine n-alkanes approximated area-weighted vegetation in the Amazon River catchment (the opposite of your finding, which highlights what makes your study interesting). Perhaps these other studies have found the same thing, or mixed results?*

- We will expand this paragraph, in order to describe briefly the results of these previous studies. This is in the discussion section of the manuscript currently, but we will move it to the introduction and expand this section.

*Lines 99–102: I suggest rewriting as "Comparison of the hillside and streambed sedimentary n-alkanes allows…". The current wording is a bit repetitive and clunky.*

- We will make this change as suggested

*Lines 104–108: I suggest splitting into two sentences, revising to: "…Tornero et al., 2016). Pleistocene sediments…"*

- We will make this change as suggested

*Lines 118–119: I suggest removing the phrase "In order to extract n-alkanes" as it is repeating the end of the previous sentence.*

- We will make this change as suggested

*Lines 120–121: please describe the solvent schedule, water content of the silica gel, and quantity of silica gel used for chromatographic separation.*

*Line 123: please describe any internal and external standards used for sample quantification. Please clarify whether and how the different response factors of the $C_{25} - C_{33}$n-alkanes were accounted for during sample quantification or in calculation of the OEP and ACL values. This is important because, while the peak area of each compound is proportional to concentration, that*

*relationship (response factor) is mass-dependent and so is slightly different for each of the n-alkanes.*

- For this study, we followed previously published extraction and quantification protocols from Brittingham et al 2017 (*Organic Geochemistry*), Brittingham et al 2019 (*Scientific Reports*) and Smolen and Hren 2023 (*Chemical Geology*). We will include the details on solvents used for separation and quantification, as well as references to these papers.
- "n-alkanes were separated from total liquid extract by passing samples through a column of activated silica gel (1.25 g) in baked Pasteur pipettes with 2 mL hexane (non-polar fraction), 4 mL dichloromethane (slightly polar fraction) and 4 mL methanol (polar fraction). n-alkanes were quantified through the analysis of the hexane fraction. We quantified n-alkanes using a BP-5 column (30 m × 0.25 mm i.d., 0.25 μm film thickness) with He as the carrier (1.5 ml/min). Oven temperature was set at 50 °C for 1 min, ramped to 180 °C at 12 °C/min, then ramped to 320 °C at 6 °C/min and held for 4 min."
  - Smolen, J. D., & Hren, M. T. (2023). Differential effects of clay mineralogy on thermal maturation of sedimentary n-alkanes. Chemical Geology, 634, 121572.
  - Brittingham, Alex, Michael T. Hren, Gideon Hartman, Keith N. Wilkinson, Carolina Mallol, Boris Gasparyan, and Daniel S. Adler. "Geochemical evidence for the control of fire by Middle Palaeolithic hominins." Scientific Reports 9, no. 1 (2019): 15368.

*Line 125: Did you forget to insert a citation here?*

- There should be a reference to Bush and McInerney (2013) here
  - Bush, Rosemary T., and Francesca A. McInerney. "Leaf wax n-alkane distributions in and across modern plants: Implications for paleoecology and chemotaxonomy." Geochimica et Cosmochimica Acta 117 (2013): 161-179.

*Line 136: How exactly did you correct for size effects? Also, I think you want to use the standard error of the mean, not standard deviation. In general, Polissar and D'Andrea (2011) [https://doi.org/10.1016/j.gca.2013.12.021] is an excellent guide on how to calculate uncertainties associated with leaf wax stable isotope measurements, particularly hydrogen isotopes.*

- We analyzed a stable isotope standard (MixA6 from A. Schimmelman) at a variety of sizes (5-30 V/s) in order to determine the relationship between peak size and measured hydrogen isotope values. We will report standard errors, rather than standard deviation, for the final manuscript.

**Results**

*Line 144: could you please provide some example chromatograms or histograms showing the n-alkane distributions in your samples? This could be as a supplementary figure if you prefer.*

- We will include example chromatograms from stream sediments and soil samples, both above and below treeline as a supplement to the final submission.

*Lines 144–145: I suggest reporting the range of carbon preference index (CPI) values of your samples to support this statement.*

- Both OEP and CPI calculations describe the relative distribution of odd and even n-alkanes in a sample, and these measurements will always have a linear correlation with one another. We believe that odd over even predominance (OEP) should be sufficient to describe the odd over even predominance of the samples, instead of describing a redundant measurement as well (CPI)

*Line 149: please replace "averages" with "is"*

- We will make this change as suggested

*Lines 150–151: Please replace "average" with "mean"*

- We will make this change as suggested

*Lines 151–153: Please specify what kind of statistical test was done here and the N samples or degrees of freedom. Same goes for the test mentioned in lines 155–156.*

- The statistical test performed here was a Student's t-test (n=30), we will specify this in the text

*Line 161: please replace "average" with mean*

- We will make this change as suggested

*Lines 170–171: does the word "significant" imply a statistical test was done here? If so, please specify what kind, the p-value, and N samples or degrees of freedom*

- We will remove the word "significant" in this sentence

*Lines 173–174: I see, this is what is referred to in 170–171. Please consider combining and condensing into a single sentence to clarify the language / keep the text concise.*

- With the removal of "significant" in the previous sentence this should help the reader

*Line 177: Please specify the N samples or degrees of freedom*

- We will include the number of soil samples in this test (n=30)

*Lines 178–179: Please revise to: "δD values were also more negative in stream sediment samples collected above the treeline (-175‰) than those collected below the treeline (-158‰)."*

- We will make this change as suggested

*Please also specify the statistical test, p-value, and N samples / degrees of freedom*

- We will include the number of stream sediment samples in this test (n=21)

*Line 180: Please revise to "…stream **sediment** samples…"*

- We will make this change as suggested

**Discussion**

*Lines 194–208: as said in my general comments, please move these lines to a new section of the methods that includes the equations and constants used in the mixing model. In addition, please specify where the satellite images used for your mapping came from with appropriate references.*

- The constants used in this section derive from the mean values provided in the results (L204), we will expand this section and repeat those values to clarify for the reader. We will also include here a map showing the different vegetation types present in the catchment, and the location of the treeline

*Line 209: I recommend beginning your revised section **4.2.** here. Also, I suggest replacing "this mixing model" with "our mixing model"*

- We will make this change as suggested

*Line 211: range of expected values for which measurement(s)? Please clarify.*

- In this study we measured δD, δ$^{13}$C, and ACL values. We will mention these measurements in this sentence.

*Line 212: Please reference a figure to back up this assertion*

- We will include a reference to figure 6 here

*Lines 213–214: Please also reference a figure here. Additionally, this is not really how ACL values (I assume that is what you are talking about here, please clarify) are used in practice. They tend to be thought of as a more qualitative indicator of vegetation / ecosystem composition. I would suggest rewriting this sentence to discuss that the n-alkane distributions do not show an expected "mixed" signal but are rather indistinguishable from the endmember values.*

- We thank the reviewer for this comment, which highlights the importance of this paper. ACL values are generally 'thought of' as a qualitative indicator, which we set to test in this natural catchment.

*Line 223: I suggest replacing "though" with "although"*

- We will make this change as suggested

*Lines 236–241: Please clarify the language here. I had to reread this section several times to understand what you mean.*

*Lines 245–254: this is a good idea, but the implementation here could use some improvement. More information is needed about how the lines in Figure 7 were calculated, as is the "expected" δD timeseries and the relative timing of the vegetation shift. It may be useful to expand this exercise into its own section in the Discussion with an accompanying short section in the Methods, but this is really just a suggestion.*

- We will rewrite these two paragraphs in order to highlight the importance of this study for understanding δD values in plant wax sedimentary archives.
- We will include more details on the thought experiment provided here. The expected δD values were calculated using mean growing season δD values in precipitation from the nearby meteorological station in Dilijan (published in Brittingham et al 2019) and the differences in mean ε values from the above and below treeline vegetation. The timing of the vegetation shifts in this is hypothetical, and we will indicate on figure 7 where these vegetation shifts will occur.
  - Brittingham, Alex, Zarmandukht Petrosyan, Joseph C. Hepburn, Michael P. Richards, Michael T. Hren, and Gideon Hartman. "Influence of the North Atlantic Oscillation on δD and δ18O in meteoric water in the Armenian Highland." Journal of Hydrology 575 (2019): 513-522.

*Line 251: I suggest replacing "heavily affected" with "influenced"*

- We will make this change as suggested

*Lines 255–258: I think this is better suited to the Introduction. Please see my comment regarding lines 83–86.*

- After expanding this section in the introduction, we will remove this from this section of the paper

**Conclusions**

No comments

**References**

*Line 294: why is this centered and not aligned to the left? Also, should it not be bolded and called "references"?*

- We will make this change as suggested

*In general, please check the formatting of your citations and ensure that all article titles have proper typesetting (subscripts, superscripts, Greek characters, etc.), that journal names are italicized, and check for typos.*

***Figures***

*In general, please ensure to upload 300 DPI or higher images or vector files of the images. The current images are a bit fuzzy.*

- We will make this change as suggested

*Figure 1: the points showing your sample sites in the right panel are very difficult to see. Could they be larger or a different shape? Perhaps a shape with a black border would help.*

- We will increase the size of the points, and increase the size of the black border around the shapes

*Figure 2: in the figure caption the samples are referred to as sediments while on the figure they are referred to as soils. Please make these consistent with each other. I also recommend switching to a colorblind friendly color palette (red-green is particularly hard to distinguish for many colorblind people).*

- We will change 'soil' in the figure legend to 'sediment' to be consistent with the captions on other figures.

*Figure 3: Please remove the lines. They really clutter up the figure and make it difficult to see the pattern that is very clearly evident in the data (really nice result!). Also, I recommend making this figure square to match the style of figure 2. Finally, please consider using three different symbol shapes and a colorblind-friendly color palette.*

- We will re-plot the data with difference symbols, color palette and remove the lines

*Figure 4: please also remove the lines here. Same comments as figure 3 regarding the color palette and symbol shapes. Also, please use $\delta^{13}C$ for your y-axis label for consistency with Figure 2.*

- We will re-plot the data with difference symbols, color palette and remove the lines, and change the axis label

*Figure 5: please also remove the lines here. Same comments as figure 3 regarding the color palette and symbol shapes. Also, please use δD for your y-axis label for consistency with Figure 2.*

- We will re-plot the data with difference symbols, color palette and remove the lines, and change the axis label

*Figure 6: Could this figure be made a little larger? It is very hard to read in this small format. Also, the low DPI of the image is really evident here. Please be sure to replace with a higher quality image. I also recommend switching to a colorblind friendly color palette and using different symbol shapes for the different panels.*

- We will increase the DPI of this figure and change the color palette

Figure 7: please see my comment regarding lines 245–254 of the text

**Technical Corrections**

*Line 34: Typo, shrubs is misspelled as "shurbs"*

*Line 47: Typo here, "supports" should be "supporting"*

*Line 78: Missing space between period and first word of following sentence "…(Gamarra et al., 2016).The fractionation…"*

*Line 172: Typo, "averages" should be "average"*

*Line 174: lower case delta is needed, not an upper case delta. It does not matter that this is the beginning of a sentence, the upper case delta means something different than the lower case delta and is not applicable here.*

*Lines 177–178: Typo here, sentence is divided by a random period*

- We will make the above changes as suggested

---

## Author Comment (AC2)

*This study, by Brittingham et al, investigates the influence of transport and depositional processes on sedimentary lipid biomarker records (leaf-wax n-alkanes and their Hydrogen and Carbon isotopic composition) by analyzing soil and stream sediments across a 1000 m altitude gradient spanning the closed deciduous forest, treeline ecotone and alpine meadow vegetation belts in a first-order catchment located in the Areguni Mountains, Armenia. Main results show that, while there is a major difference in the soil and stream n-alkane and their isotopic values above and below the treeline, stream sediment biomarkers below the treeline predominantly reflect local vegetation rather than upstream contributions. This finding is important for the interpretation of sediment biomarker records, in that it shows that processes at the level of the catchment must also be accounted for and may critically influence the distribution of different biomarker compounds in sediment archives.*

*The manuscript reads very well, aims are clear, the topic approached is relevant and addresses an important knowledge gap in the field of biomarker-based palaeoenvironmental and palaeoclimate reconstructions. Some further clarifications are needed in the overall design of the study and interpretation of the results (details below), but most of these are minor. The main recommendation for the authors is to describe the study area more thoroughly, in a separate section, and provide information on temperature and precipitation patterns, geology and soil types (for example, are soils acidic?) and dominant vegetation species for each of the vegetation belts, because in my opinion this information is relevant for interpreting biomarker distribution. Then, I think it is important to show at least some of the more representative chromatograms (these may be even placed in the supplementary material). Also, there is an issue that, in my opinion, needs to be further expanded in the discussion: if the top 10 cm of soil were removed prior to subsampling for biomarker analysis (L. 116), this means that the collected soil samples likely do not represent modern vegetation, as soil takes a long time to form. Conversely, stream bed sediments may be of a more recent age compared to the soil samples, as stream beds are highly dynamic environments. How could this potential age discrepancy impact the results?*

*I therefore recommend the manuscript for publication, provided that these clarifications are addressed.*

- We thank the reviewer for their helpful comments on the draft of this paper. Detailed responses to the points addressed by reviewer 2 can be found below.

Minor comments

*Title: could be adjusted a bit, because it suggests a generalized conclusion, while the results are study-case based, and it is not clear to what extent these findings can be extrapolated to all high-altitude catchments.*

- We will alter the title as follows: "Locally Produced Leaf Wax Biomarkers in the High-Altitude Areguni Mountains Outweigh Downstream Transport"

Introduction:

*L.45-48 A word is missing from this sentence: "and Iran, 'that' supports a wide variety…"?*

- We will make this change as suggested

*L.49-54 The link between interpretation of palaeoclimatic records in the study region and the environmental signal of biomarkers in sedimentary archives is unclear. I suggest adding a sentence to explain why understanding processes involved in the sedimentary integration of biomarkers, and the scale of biomarker environmental signals are relevant for palaeoclimatic reconstructions.*

- As a similar suggestion was also made by reviewer 1, we will add a section referencing previous biomarker work done in the region to highlight the importance of this work for paleoclimate reconstruction

*L 64-72 Why are the carbon isotope values of C3 vs C4 plants relevant for the study area or for the aims of this study? E.g., were there major shifts in the importance of C3 vs C4 plants in the vegetation history of the area? Is the proportion of the C4 plants in the current vegetation increasing?*

- This area does have a vegetation history which includes C4 vegetation in the Holocene (Tornero et al 2016), and roughly 3% of plant species in Armenia are C4 today (Rudov et al 2020). We will add references to these publications in the revised manuscript.
    - Rudov, Alexander, Marjan Mashkour, Morteza Djamali, and Hossein Akhani. "A review of C4 plants in southwest Asia: an ecological, geographical and taxonomical analysis of a region with high diversity of C4 eudicots." Frontiers in plant science 11 (2020): 546518.
    - Tornero, Carlos, Marie Balasse, Adrian Bălăşescu, Christine Chataigner, Boris Gasparyan, and Cyril Montoya. "The altitudinal mobility of wild sheep at the Epigravettian site of Kalavan 1 (Lesser Caucasus, Armenia): Evidence from a sequential isotopic analysis in tooth enamel." Journal of Human Evolution 97 (2016): 27-36.

*L.73 Same as in the previous comment, it is hard to grasp the relevance of hydrogen isotope values in leaf-wax n-alkanes for the study area or for the aims of this study.*

- Since we include hydrogen isotope value measurements in this study, we believe that it is important to include some background information on what influences those values. Primarily the physiological drivers of fractionation by different plant

species, which we observe in this study with the above and below treeline vegetation.

*L.83-86 Could you, please, summarize some of the key findings of the referenced publications, that are also relevant for this study?*

- We briefly describe the key findings of these referenced publications later in the manuscript (L255-L258), however at the suggestion of both reviewers we will include a larger paragraph in the introduction to summarize the findings of these papers.

*L.105 What proxies were used to assess the relationships between the past treeline and climate? Are there knowledge gaps that remained unaddressed and that are addressed within the present study?*

- These previous studies included analysis of pollen, carbon and oxygen isotope values of herbivore tooth enamel, and biomarkers. This study is designed to address some of the interpretive gaps in Malinsky-Buller et al, 2021, in which we analyzed biomarkers in fluvial sediments deposited between 60-45 ka in this catchment.

*L.108 Regarding the potential of sediments at Kalavan to reconstruct the treeline-climate relationship, I assume it refers to a biomarker-based reconstruction, because it is not clear. But then, why would biomarkers be the preferred proxy instead of more established proxies, like for example plant macro-remains and pollen? Justification needs to be a bit stronger here.*

- We will make it more clear that we are referring specifically to biomarker integration processes. Plant macro-remains and pollen also are likely to be subject to uncertainty in their transportation processes, which remain unstudied in this environment, and therefore we believe biomarkers will provide a good proxy for treeline relationship in the Pleistocene sediments present in this region.

Methods:

*I recommend the authors to begin with a subsection which describes the study area in terms of climate, geology, soil types and dominant vegetation species for the two main vegetation belts and the treeline ecotone. I also recommend the authors to create another section, that could be placed last, that collates the description of the statistical methods used (significance tests and mixture models).*

- We will expand the background information provided in the methods section. Descriptions of the dominant vegetation in the area from Joannin et al 2022 and Volodicheva 2002 and the geological background from Malinsky-Buller et al 2022, 2024.

- Volodicheva, Natalya. "The Caucasus." The physical geography of northern Eurasia (2002): 350-376.
- Joannin, Sébastien, A. Capit, V. Ollivier, O. Bellier, B. Brossier, B. Mourier, P. Tozalakian et al. "First pollen record from the Late Holocene forest environment in the Lesser Caucasus." Review of Palaeobotany and Palynology 304 (2022): 104713.
- Malinsky-Buller, Ariel, Philip Glauberman, Vincent Ollivier, Tobias Lauer, Rhys Timms, Ellery Frahm, Alexander Brittingham et al. "Short-term occupations at high elevation during the Middle Paleolithic at Kalavan 2 (Republic of Armenia)." PLoS One 16, no. 2 (2021): e0245700.
- Malinsky-Buller, Ariel, Lotan Edeltin, Vincent Ollivier, Sébastien Joannin, Odile Peyron, Tobias Lauer, Ellery Frahm et al. "The environmental and cultural background for the reoccupation of the Armenian Highlands after the Last Glacial Maximum: The contribution of Kalavan 6." Journal of Archaeological Science: Reports 56 (2024): 104540.

*L.119-120 Please provide a reference for the Soxhlet procedure used for lipid extraction. What intrigues me is the relatively high proportion of methanol in the solvent mixture and the long extraction time.*

*L .121 Please specify what solvent or solvent mixture was used for n-alkane separation.*

*L.121-125 Please add details on: oven temperature, use of blanks to test for lab contamination, standards used for n-alkane quantification, method used for integration of peak areas etc.*

- For this study, we followed previously published extraction and quantification protocols from Brittingham et al 2017 (*Organic Geochemistry*), Brittingham et al 2019 (*Scientific Reports*) and Smolen and Hren 2023 (*Chemical Geology*). We will include the details on solvents used for separation and quantification, as well as references to these papers.
- "n-alkanes were separated from total liquid extract by passing samples through a column of activated silica gel (1.25 g) in baked Pasteur pipettes with 2 mL hexane (non-polar fraction), 4 mL dichloromethane (slightly polar fraction) and 4 mL methanol (polar fraction). n-alkanes were quantified through the analysis of the hexane fraction. We quantified n-alkanes using a BP-5 column (30 m × 0.25 mm i.d., 0.25 μm film thickness) with He as the carrier (1.5 ml/min). Oven temperature was set at 50 °C for 1 min, ramped to 180 °C at 12 °C/min, then ramped to 320 °C at 6 °C/min and held for 4 min."
  - Smolen, J. D., & Hren, M. T. (2023). Differential effects of clay mineralogy on thermal maturation of sedimentary n-alkanes. Chemical Geology, 634, 121572.
  - Brittingham, Alex, Michael T. Hren, Gideon Hartman, Keith N. Wilkinson, Carolina Mallol, Boris Gasparyan, and Daniel S. Adler. "Geochemical evidence for the control of fire by Middle Palaeolithic hominins." Scientific Reports 9, no. 1 (2019): 15368.

*L.125 'REF' shows a missing reference?*

- There should be a reference to Bush and McInerney (2013) here
    - o Bush, Rosemary T., and Francesca A. McInerney. "Leaf wax n-alkane distributions in and across modern plants: Implications for paleoecology and chemotaxonomy." Geochimica et Cosmochimica Acta 117 (2013): 161-179.

Results:

*L.143-145 It would be great to see some of the most illustrative chromatograms added to the supplementary file. This would help the reader understand better the n-alkane distribution in different sets of samples.*

- We will add illustrative chromatograms in the supplementary material

*L.174 Please, add the design of the significance test to the methods section.*

- We used student's t-tests for our significance tests, we will include this in the revised draft of this paper.

*L.152 word missing: 'between the average values of the "n-alkane?" above treeline and below...'*

- We will make this change as recommended and add "n-alkane" in this section

*I don't see any description of results obtained for the mixing model.*

- In the revised submission, we will include results and tables from the mixing model. This will include sample and vegetation composition of the stream samples, and the endmember constants for each of the variables.

*Discussion:*

*L.184-186 Please, reference the relevant figures here.*

- We will increase the references to relevant figures, both here and throughout the manuscript

*L.194-208 Consider moving these paragraphs in a separate section of the methods, where you could also include information about the statistical tests used. But overall, I very much like the idea of using a mixing model to compare expected and obtained biomarker compound values.*

- As per recommendations from both of the reviewers, we will expand this section to include more details of the mixing model. This will include sample and vegetation composition of the stream samples, and the endmember constants for each of the variables.

*L.200 What does the phrase 'tree and grass sediment' refer to?*

- Here we refer to the areas covered by forest and grassland. We will change the wording of this sentences for clarity.

*L.210-212 Please reference the relevant figure for this statement.*

- We will add references to figure 6 here

*L.212 I assume 'deciduous-sourced n-alkanes' refers to deciduous trees, but it's a bit ambiguous, because there are also deciduous herbaceous plants. Could you also include what the dominant deciduous species in the forest are?*

- In order to make this clearer to the reader, we will refer to these as "n-alkanes sourced from below-treeline vegetation". We will add a section in the background describing the vegetation structure and dominant vegetation types in the forest (Oak/Beech/Hornbeam)

*L.228 Please include the distance from the study site of the lacustrine core that was analyzed for pollen.*

- The lacustrine core described here is ~5 km from the study area, at an altitude of 1912 meters. We will add this information to the manuscript.

*L 233-234 This general statement needs a reference.*

- This statement is in reference to this study. We will re-word this sentence to make it clear that we are referring to the specific results from ACL values and hydrogen and carbon isotopes in this study.

*L.261-263 As an additional research direction, perhaps collecting water samples for lipid analysis could help clarifying the role of transport and depositional processes.*

- We appreciate the suggestion from the reviewer, this would be an interesting direction for future research

*Figures: most of them are blurry and should be uploaded in a better resolution.*

- We will re-upload the figures in higher resolution

*Fig. 1 Could you, please, specify the source of the satellite images? Also, I would find it more relevant if the figure included a close-up of the studied catchment with sampling points superposed on vegetation types.*

- We will include a more zoomed in map of the sampling locations, so that the sample points are easier to distinguish.

*Figures 3-5. I assume the green and red rectangles are soil samples (although it is not clear, and also not colorblind-friendly). But it should also be clarified which of the stream samples (blue triangles) were taken from above and from below the treeline respectively.*

- Though we refer to these in the figure caption, we will also include the 'sediment' label in the legend. We will also change the color of these symbols.

*Figure 6. Please make it larger, and also increase the resolution, because the labels are hardly visible.*
- We will make this change as recommended

---

## Author Response (AR1)

**Reviewer 1**

**General comments:**

*My main comment on the manuscript is a request that the authors include information about their mixing model, including necessary equations and values used for constants, as a subsection in the methods. The values for constants could be added as a table or in the text. The additional text and equations would substantially clarify the discussion in section **4.2.** and, more importantly, are necessary for evaluating the comparison between measured and expected values shown in Figure 6. Much of the necessary text can be moved from lines 194–208 to the methods section.*

- We have added a table with constants (Table 1) to further clarify the mixing model used to calculated expected stream values. We have also expanded the text in section 4.2 to further clarify the results from this data.

*I also strongly recommend appending the supporting data for this study as supplementary tables or as files uploaded to a publicly accessible database. The data may prove useful for future paleoclimate or palaeoecological work in this region, so it is important to make sure the data is accessible to those who may need it in the future.*

- We have added a supplementary Excel file including all of this data.

*Lastly, I recommend inserting more references to the figures within the text. As is, each figure is only referenced once or twice, even in sections where the data is discussed in detail. More figure references will make the text easier to follow.*

- We added further references to figures to help the reader follow text references (L204, 244, 250, 255)

**Specific comments**

*Title*

*I suggest amending the title to better reflect the findings and conclusions of the study. As is, the title appears to suggest that the findings here are applicable to all high-altitude river (and lake?) catchments. However, this assertion is not made in the manuscript.*

*Two other things to point: (1) n-alkanes measured in this study should really be referred to as leaf wax or vegetation biomarkers since they are produced by plants, not produced by sediments; (2) I believe you mean downstream river transport, as in away from the river's headwaters. As written, the word upstream implies transport of sediment in the opposite direction of the water flow, which I do not think is what you mean.*

*A potential alternative title could be:*

*"Locally Produced Leaf Wax Biomarkers in the High-Altitude Lesser Caucuses Outweigh Downstream Transport"*

- The new title of the manuscript is "Locally Produced Leaf Wax Biomarkers in the High-Altitude Areguni Mountains Outweigh Downstream Transport"

***Abstract***

*Lines 18–22: these first two sentences are a bit repetitive. I suggest removing the first sentence that reads, "Sedimentary records…signals." and starting the abstract with the second sentence that reads "The integration of…". This still introduces the study effectively while making the language more succinct.*

- This section of the abstract now reads (L18-21) "Sedimentary records of lipid biomarkers such as leaf wax n-alkanes are not only influenced by ecosystem turnover and physiological changes in plants, they are also influenced by earth surface processes integrating these signals into the sedimentary record, though the effect of these integration processes are not fully understood."

*Line 25: I suggest removing "the existence of" and just stating "We utilize a treeline…"*

- We made this change

*Lines 35–36: Please clarify the language here, I am unsure what you mean by the "latter observation" since the language in lines 32–35 makes it seem like you do not have sufficient evidence to distinguish which mechanism is driving the pattern you observe in the stream sediments (this is also what I took away from reading the rest of the manuscript).*

- This line now reads begins (L34): "Though these observations may preclude using *n*-alkanes to measure past treeline movement in these mountains"

*Line 38: I am unclear what "changes in upstream fractionation differences" means here. I suspect you are talking about fractionation of leaf wax hydrogen isotope composition caused by changes in landscape vegetation, please clarify the language as this confused me a bit.*

- We changed this text to (L36): "δD values of biomarkers in fluvial deposits in these settings are more likely to record local hydrological changes rather than reflect fractionation changes due to turnover in upstream vegetation structure.".

***Introduction***

*Line 45: Is "so called" necessary here?*

- We removed this

*Line 49: I suggest adding a sentence here briefly summarizing any paleoclimate records from this region that utilize leaf wax n-alkanes if there are any as that will clarify the importance of this study to your readers.*

- We added references to the following papers, which to our knowledge are the only ones that have used plant waxes in Armenia.
- L50: "Plant wax biomarkers have been used in this region in both geological and archaeological contexts to reconstruct past climates, therefore understanding modern variability and transport processes will help refine these interpretations (Brittingham et al., 2019; Glauberman et al., 2020; Malinsky-Buller et al., 2021, 2024; Trigui et al., 2019)."
    - Brittingham, Alex, Michael T. Hren, Gideon Hartman, Keith N. Wilkinson, Carolina Mallol, Boris Gasparyan, and Daniel S. Adler. "Geochemical evidence for the control of fire by Middle Palaeolithic hominins." *Scientific Reports* 9, no. 1 (2019): 15368.
    - Malinsky-Buller, Ariel, Philip Glauberman, Vincent Ollivier, Tobias Lauer, Rhys Timms, Ellery Frahm, Alexander Brittingham et al. "Short-term occupations at high elevation during the Middle Paleolithic at Kalavan 2 (Republic of Armenia)." *PLoS One* 16, no. 2 (2021): e0245700.
    - Glauberman, Phil, Boris Gasparyan, Jennifer Sherriff, Keith Wilkinson, Bo Li, Monika Knul, Alex Brittingham et al. "Barozh 12: Formation processes of a late Middle Paleolithic open-air site in western Armenia." *Quaternary Science Reviews* 236 (2020): 106276.
    - Malinsky-Buller, Ariel, Lotan Edeltin, Vincent Ollivier, Sébastien Joannin, Odile Peyron, Tobias Lauer, Ellery Frahm et al. "The environmental and cultural background for the reoccupation of the Armenian Highlands after the Last Glacial Maximum: The contribution of Kalavan 6." *Journal of Archaeological Science: Reports* 56 (2024): 104540.
    - Trigui, Yesmine, Daniel Wolf, Lilit Sahakyan, Hayk Hovakimyan, Kristina Sahakyan, Roland Zech, Markus Fuchs, Tilmann Wolpert, Michael Zech, and Dominik Faust. "First calibration and application of leaf wax n-alkane biomarkers in loess-paleosol sequences and modern plants and soils in Armenia." *Geosciences* 9, no. 6 (2019): 263.

*Line 62: I suggest replacing "though" with "although"*

- Made this change as suggested

*Line 64: I suggest rewriting as "The carbon isotope ($\delta^{13}C$) composition of plant tissue is primarily set by the photosynthetic pathway of the plant.*

- Made this change as suggested

*Line 72: Could you please comment on which of these factors is important for your study site? For example, are there $C_4$ plants in the Caucuses?*

- We have added the following sentence and references:
- "Currently, C4 vegetation makes up around 3% of plant species in Armenia (Rudov et al., 2020), and was present in the Kalavan region during the Holocene (Tornero et al., 2016)."
  - Rudov, Alexander, Marjan Mashkour, Morteza Djamali, and Hossein Akhani. "A review of C4 plants in southwest Asia: an ecological, geographical and taxonomical analysis of a region with high diversity of C4 eudicots." Frontiers in plant science 11 (2020): 546518.
  - Tornero, Carlos, Marie Balasse, Adrian Bălăşescu, Christine Chataigner, Boris Gasparyan, and Cyril Montoya. "The altitudinal mobility of wild sheep at the Epigravettian site of Kalavan 1 (Lesser Caucasus, Armenia): Evidence from a sequential isotopic analysis in tooth enamel." Journal of Human Evolution 97 (2016): 27-36.

*Lines 83–86: I suggest rewriting this sentence to summarize the common findings of the studies you cite here. For example, I recall that the Feakins et al. 2018a study cited here found that riverine n-alkanes approximated area-weighted vegetation in the Amazon River catchment (the opposite of your finding, which highlights what makes your study interesting). Perhaps these other studies have found the same thing, or mixed results?*

- We have changed the following text to further clarify the results from previous integrations studies (L90-97):
- "Previous studies on the integration of organic biomarkers have produced mix results, with some demonstrating spatial integration of catchment signals (Alewell et al., 2016; Feakins et al., 2018; Hemingway et al., 2016), whereas others did not observe this (Häggi et al., 2016; Ponton et al., 2014). However, these previous studies typically focused on very large river systems, which will undergo different transport processes than the first-order streams analyzed in this study. A number of these studies (Alewell et al., 2016; Feakins et al., 2018; Hemingway et al., 2016; Ponton et al., 2014) also observed seasonal differences in biomarker load in river sediments."

*Lines 99–102: I suggest rewriting as "Comparison of the hillside and streambed sedimentary n-alkanes allows…". The current wording is a bit repetitive and clunky.*

- We made this change as suggested

*Lines 104–108: I suggest splitting into two sentences, revising to: "…Tornero et al., 2016). Pleistocene sediments…"*

- We made this change as suggested

*Lines 118–119: I suggest removing the phrase "In order to extract n-alkanes" as it is repeating the end of the previous sentence.*

- We made this change as suggested

*Lines 120–121: please describe the solvent schedule, water content of the silica gel, and quantity of silica gel used for chromatographic separation.*

*Line 123: please describe any internal and external standards used for sample quantification. Please clarify whether and how the different response factors of the $C_{25} – C_{33}$ n-alkanes were accounted for during sample quantification or in calculation of the OEP and ACL values. This is important because, while the peak area of each compound is proportional to concentration, that relationship (response factor) is mass-dependent and so is slightly different for each of the n-alkanes.*

- We have expanded the description of our extraction methods (L132-140):
- "Samples were extracted using a Soxhlet apparatus with 2:1 dichloromethane:methanol for 48 hours. Following lipid extraction, n-alkanes were separated from total liquid extract by passing samples through a column of activated silica gel (1.25 g) in baked Pasteur pipettes with 2 mL hexane (non-polar fraction), 4 mL dichloromethane (slightly polar fraction) and 4 mL methanol (polar fraction). n-alkanes were quantified through the analysis of the hexane fraction. We quantified n-alkanes using a BP-5 column (30 m × 0.25 mm i.d., 0.25 μm film thickness) with He as the carrier (1.5 ml/min). Oven temperature was set at 50 °C for 1 min, ramped to 180 °C at 12 °C/min, then ramped to 320 °C at 6 °C/min and held for 4 min. (Brittingham et al., 2017; Smolen and Hren, 2023). ramped to 320 °C at 6 °C/min and held for 4 min. (Brittingham et al., 2017; Smolen and Hren, 2023). We measured a standard mixture of *n*-alkanes ($C_{20}$-$C_{33}$) of known concentration to correct for mass dependent response decreases in longer chain *n*-alkanes."
    - Smolen, J. D., & Hren, M. T. (2023). Differential effects of clay mineralogy on thermal maturation of sedimentary n-alkanes. Chemical Geology, 634, 121572.
    - Brittingham, Alex, Michael T. Hren, Gideon Hartman, Keith N. Wilkinson, Carolina Mallol, Boris Gasparyan, and Daniel S. Adler. "Geochemical evidence for the control of fire by Middle Palaeolithic hominins." Scientific Reports 9, no. 1 (2019): 15368.

*Line 125: Did you forget to insert a citation here?*

- We corrected this, and added the following citation:
    - Bush, Rosemary T., and Francesca A. McInerney. "Leaf wax n-alkane distributions in and across modern plants: Implications for paleoecology and chemotaxonomy." Geochimica et Cosmochimica Acta 117 (2013): 161-179.

*Line 136: How exactly did you correct for size effects? Also, I think you want to use the standard error of the mean, not standard deviation. In general, Polissar and D'Andrea (2011) [https://doi.org/10.1016/j.gca.2013.12.021] is an excellent guide on how to calculate*

*uncertainties associated with leaf wax stable isotope measurements, particularly hydrogen isotopes.*

- We analyzed a stable isotope standard (MixA6 from A. Schimmelman) at a variety of sizes (5-30 V/s) in order to determine the relationship between peak size and measured hydrogen isotope values. We report standard errors rather than standard deviations:
- L153-156: "Internal standards (Mix A5 from A. Schimmelman) were run every four samples across a range of concentrations (5-30 V/s) to correct for size effects. Standard errors were 0.4‰ for δ13C and 3‰ for δD. Isotope ratios (R) were converted to δX (δ13C and δD) values (Eq. 3) and are expressed in permill (‰).

**Results**

*Line 144: could you please provide some example chromatograms or histograms showing the n-alkane distributions in your samples? This could be as a supplementary figure if you prefer.*

- We have added a supplemental file which includes 4 chromatograms, one for each type of sample analyzed (Above treeline soil, below treeline soil, above treeline stream, and below treeline stream)

*Lines 144–145: I suggest reporting the range of carbon preference index (CPI) values of your samples to support this statement.*

- Both OEP and CPI calculations describe the relative distribution of odd and even n-alkanes in a sample, and these measurements will always have a linear correlation with one another. We believe that odd over even predominance (OEP) should be sufficient to describe the odd over even predominance of the samples, instead of describing a redundant measurement as well (CPI)

*Line 149: please replace "averages" with "is"*

- We made this change as suggested

*Lines 150–151: Please replace "average" with "mean"*

- We made this change as suggested

*Lines 151–153: Please specify what kind of statistical test was done here and the N samples or degrees of freedom. Same goes for the test mentioned in lines 155–156.*

- We added a reference to the number of samples in the text:
- L170-173: ). In soils above the treeline, the mean ACL value is 30.6 (range of 29.8-31.8). In soils below the treeline, the mean ACL value is 29.5 (range of 28.4-30.4). There is a significant difference between the average ACL values of the nalkanes in above treeline and below treeline soils (Student's t-test, p<0.001, n=30). Stream sediment above the treeline have an average ACL value of 29.7 (range of 29.1-30.2) and stream sediments below the treeline have an average ACL value of 29.3 (range of 28.6 -30.0). The stream sediments from below the treeline have a significantly (Student's t-test, p<0.001, n=21) lower average ACL value than those above the treeline."

*Line 161: please replace "average" with mean*

- We made this change as suggested

*Lines 170–171: does the word "significant" imply a statistical test was done here? If so, please specify what kind, the p-value, and N samples or degrees of freedom*

- We made this change as suggested

*Lines 173–174: I see, this is what is referred to in 170–171. Please consider combining and condensing into a single sentence to clarify the language / keep the text concise.*

- This section now reads:
- L180-182: "However, we do not find a difference between the Paq values in the stream sediments when compared to the soil samples, indicating that the organic load of the stream sediments is mostly of terrestrial origin. Terrestrial plants have average Paqvalues of 0.09, with emergent plants averaging 0.25 (Ficken et al., 2000). Only eight of the 51 samples in this study had Paqvalues above 0.20, four stream and four soil samples."

*Line 177: Please specify the N samples or degrees of freedom*

- We included the n samples here, this text now reads:
- L193-195: Above the treeline, $\delta$13C values in soils average -34.9‰, and below the treeline soil alkanes average -33.3‰ (p<0.0001, student's t-test, n=30). Stream sediment $\delta$13C values average -35.0‰ above the treeline and -33.6‰ below the treeline  (p<0.0001, student's t-test, n=21).

*Lines 178–179: Please revise to: "$\delta$D values were also more negative in stream sediment samples collected above the treeline (-175‰) than those collected below the treeline (-158‰)."*

*Please also specify the statistical test, p-value, and N samples / degrees of freedom*

- We made this change as suggest, this section is now as follows:
- L198-202: These values were significantly more negative in above treeline sediments (-175‰) than in below treeline sediments (-156‰) (p<0.001, student's t-test, n=30). $\delta$D values were also more negative in stream sediment samples collected above the treeline (-175‰) than below the treeline (-158‰) (p<0.001,

student's t-test, n=21).  As with the δ13C values, the δD values of stream sediment samples show sudden change as the stream drops below the treeline.

*Line 180: Please revise to "…stream **sediment** samples…"*

- We made this change as suggested

**Discussion**

*Lines 194–208: as said in my general comments, please move these lines to a new section of the methods that includes the equations and constants used in the mixing model. In addition, please specify where the satellite images used for your mapping came from with appropriate references.*

- We have expanded this section to further describe the results from the mixing model. We have included a new figure showing the area-based model (Fig. 6) as well as a table (Table 1) to provide the constants used. This section now reads as follows:
- L220-233 "The parameters we used for our mixing model are: 1. Satellite images (Google Earth) to map the areas covered by alpine meadow and forest vegetation throughout the Dany River catchment. 2. An estimate of net primary productivity of organic material production in grasses and trees (grams per area) (Brun et al., 2022). 3. Estimates of n-alkane production in grasses and trees in the Greater and Lesser Caucasus Mountains (grams of n-alkane per gram of organic material) (Bliedtner et al., 2018; Trigui et al., 2019). 4. End member values of δD, δ13C and ACL derived from the average hillslope soils above and below the treeline. At each sample point within the catchment, we first calculated the upstream area covered by the two dominant vegetation types within the catchment (deciduous forest and alpine meadow) (Figure 6). This area was then multiplied by the previously mentioned constants (Table 1). By multiplying these terms (area x organic mass production x n-alkane production x end member soils value), we created an n-alkane production map for the Dany River catchment. Using this method, we calculated, the amount of grass and tree n-alkanes produced on the hillslopes above the sampling locations and the expected δD, δ13C and ACL values for each stream sampling location (Figure 7a, 7c, 7e)."

*Line 209: I recommend beginning your revised section **4.2.** here. Also, I suggest replacing "this mixing model" with "our mixing model"*

- We replaced "this" with "our" (L246)

*Line 211: range of expected values for which measurement(s)? Please clarify.*

*Line 212: Please reference a figure to back up this assertion*

- L247 now reads "Measured δD, δ13C and ACL values do not have a linear relationship with the expected values based on vegetation area (Fig 7b, 7d, 7f)."

*Lines 213–214: Please also reference a figure here. Additionally, this is not really how ACL values (I assume that is what you are talking about here, please clarify) are used in practice. They tend to be thought of as a more qualitative indicator of vegetation / ecosystem composition. I would suggest rewriting this sentence to discuss that the n-alkane distributions do not show an expected "mixed" signal but are rather indistinguishable from the endmember values.*

- We have added a reference to Fig 7 here.

*Line 223: I suggest replacing "though" with "although"*

- We have made this change as suggested

*Lines 236–241: Please clarify the language here. I had to reread this section several times to understand what you mean.*

*Lines 245–254: this is a good idea, but the implementation here could use some improvement. More information is needed about how the lines in Figure 7 were calculated, as is the "expected" δD timeseries and the relative timing of the vegetation shift. It may be useful to expand this exercise into its own section in the Discussion with an accompanying short section in the Methods, but this is really just a suggestion.*

- We have added references to the source of hydrogen isotope value time series here. The expected δD values were calculated using mean growing season δD values in precipitation from the nearby meteorological station in Dilijan (published in Brittingham et al 2019) and the differences in mean ε values from the above and below treeline vegetation. The timing of the vegetation shifts in this is hypothetical, which we have noted in the text.
  - Brittingham, Alex, Zarmandukht Petrosyan, Joseph C. Hepburn, Michael P. Richards, Michael T. Hren, and Gideon Hartman. "Influence of the North Atlantic Oscillation on δD and δ18O in meteoric water in the Armenian Highland." Journal of Hydrology 575 (2019): 513-522.

*Line 251: I suggest replacing "heavily affected" with "influenced"*

- W have made this change as suggested

*Lines 255–258: I think this is better suited to the Introduction. Please see my comment regarding lines 83–86.*

- We have removed this section from here and moved it to the introduction

**Conclusions**

No comments

**References**

*Line 294: why is this centered and not aligned to the left? Also, should it not be bolded and called "references"?*

- We have made this change as suggested

*In general, please check the formatting of your citations and ensure that all article titles have proper typesetting (subscripts, superscripts, Greek characters, etc.), that journal names are italicized, and check for typos.*

***Figures***

*In general, please ensure to upload 300 DPI or higher images or vector files of the images. The current images are a bit fuzzy.*

- We have reuploaded the figures in higher resolution

*Figure 1: the points showing your sample sites in the right panel are very difficult to see. Could they be larger or a different shape? Perhaps a shape with a black border would help.*

*Figure 2: in the figure caption the samples are referred to as sediments while on the figure they are referred to as soils. Please make these consistent with each other. I also recommend switching to a colorblind friendly color palette (red-green is particularly hard to distinguish for many colorblind people).*

- We have changed the color palette to yellow-green in order to match the new figure showing the areas covered by these different types of vegetation.

*Figure 3: Please remove the lines. They really clutter up the figure and make it difficult to see the pattern that is very clearly evident in the data (really nice result!). Also, I recommend making this figure square to match the style of figure 2. Finally, please consider using three different symbol shapes and a colorblind-friendly color palette.*

*Figure 4: please also remove the lines here. Same comments as figure 3 regarding the color palette and symbol shapes. Also, please use $\delta^{13}C$ for your y-axis label for consistency with Figure 2.*

*Figure 5: please also remove the lines here. Same comments as figure 3 regarding the color palette and symbol shapes. Also, please use $\delta D$ for your y-axis label for consistency with Figure 2.*

- In order to de-clutter this figure, we have removed the stream sediment samples from these three plots, and changed the color palette. These samples are plotted along their elevation gradient in Figure 8, this should make the data visually clearer.

**Technical Corrections**

*Line 34: Typo, shrubs is misspelled as "shurbs"*

*Line 47: Typo here, "supports" should be "supporting"*

*Line 78: Missing space between period and first word of following sentence "…(Gamarra et al., 2016).The fractionation…"*

*Line 172: Typo, "averages" should be "average"*

*Line 174: lower case delta is needed, not an upper case delta. It does not matter that this is the beginning of a sentence, the upper case delta means something different than the lower case delta and is not applicable here.*

*Lines 177–178: Typo here, sentence is divided by a random period*

- We have made the above changes as suggested

**Reviewer 2**

*This study, by Brittingham et al, investigates the influence of transport and depositional processes on sedimentary lipid biomarker records (leaf-wax n-alkanes and their Hydrogen and Carbon isotopic composition) by analyzing soil and stream sediments across a 1000 m altitude gradient spanning the closed deciduous forest, treeline ecotone and alpine meadow vegetation belts in a first-order catchment located in the Areguni Mountains, Armenia. Main results show that, while there is a major difference in the soil and stream n-alkane and their isotopic values above and below the treeline, stream sediment biomarkers below the treeline predominantly reflect local vegetation rather than upstream contributions. This finding is important for the interpretation of sediment biomarker records, in that it shows that processes at the level of the catchment must also be accounted for and may critically influence the distribution of different biomarker compounds in sediment archives.*

*The manuscript reads very well, aims are clear, the topic approached is relevant and addresses an important knowledge gap in the field of biomarker-based palaeoenvironmental and palaeoclimate reconstructions. Some further clarifications are needed in the overall design of the study and interpretation of the results (details below), but most of these are minor. The main recommendation for the authors is to describe the study area more thoroughly, in a separate section, and provide information on temperature and precipitation patterns, geology and soil types (for example, are soils acidic?) and dominant vegetation species for each of the vegetation belts, because in my opinion this information is relevant for interpreting biomarker distribution. Then, I think it is important to show at least some of the more representative chromatograms (these may be even placed in the supplementary material). Also, there is an issue that, in my opinion, needs to be further expanded in the discussion: if the top 10 cm of soil were removed prior to subsampling for biomarker analysis (L. 116), this means that the collected soil samples likely do not represent modern vegetation, as soil takes a long time to form. Conversely, stream bed sediments may be of a more recent age compared to the soil samples, as stream beds are highly dynamic environments. How could this potential age discrepancy impact the results?*

*I therefore recommend the manuscript for publication, provided that these clarifications are addressed.*

- We thank the reviewer for their helpful comments on the draft of this paper. Detailed responses to the points addressed by reviewer 2 can be found below.

Minor comments

*Title: could be adjusted a bit, because it suggests a generalized conclusion, while the results are study-case based, and it is not clear to what extent these findings can be extrapolated to all high-altitude catchments.*

- The new title of the manuscript is "Locally Produced Leaf Wax Biomarkers in the High-Altitude Areguni Mountains Outweigh Downstream Transport"

Introduction:

*L.45-48 A word is missing from this sentence: "and Iran, 'that' supports a wide variety…"?*

- We have made this change, this sentence is now: (L44-46) "The Caucasus Region in particular has been identified as a biodiversity hotspot covering the Republics of Armenia, Georgia, Azerbaijan, and parts of the Russian Federation, Türkiye, and Iran, that supports a wide variety of plant and animal species (Zazanashvili, 2009; Gasparyan and Glauberman, 2022)."

*L.49-54 The link between interpretation of palaeoclimatic records in the study region and the environmental signal of biomarkers in sedimentary archives is unclear. I suggest adding a sentence to explain why understanding processes involved in the sedimentary integration of biomarkers, and the scale of biomarker environmental signals are relevant for palaeoclimatic reconstructions.*

- We have added references to paleoclimate studies in the region which have used biomarkers, and this now reads (L50-53):
- "Plant wax biomarkers have been used in this region in both geological and archaeological contexts to reconstruct past climates, therefore understanding modern variability and transport processes will help refine these interpretations (Brittingham et al., 2019; Glauberman et al., 2020; Malinsky-Buller et al., 2021, 2024; Trigui et al., 2019)."

*L 64-72 Why are the carbon isotope values of C3 vs C4 plants relevant for the study area or for the aims of this study? E.g., were there major shifts in the importance of C3 vs C4 plants in the vegetation history of the area? Is the proportion of the C4 plants in the current vegetation increasing?*

- We have added the following sentence and references:
- "Currently, C4 vegetation makes up around 3% of plant species in Armenia (Rudov et al., 2020), and was present in the Kalavan region during the Holocene (Tornero et al., 2016)."
  - Rudov, Alexander, Marjan Mashkour, Morteza Djamali, and Hossein Akhani. "A review of C4 plants in southwest Asia: an ecological, geographical and taxonomical analysis of a region with high diversity of C4 eudicots." Frontiers in plant science 11 (2020): 546518.
  - Tornero, Carlos, Marie Balasse, Adrian Bălăşescu, Christine Chataigner, Boris Gasparyan, and Cyril Montoya. "The altitudinal mobility of wild sheep at the Epigravettian site of Kalavan 1 (Lesser Caucasus, Armenia): Evidence from a sequential isotopic analysis in tooth enamel." Journal of Human Evolution 97 (2016): 27-36.

*L.73 Same as in the previous comment, it is hard to grasp the relevance of hydrogen isotope values in leaf-wax n-alkanes for the study area or for the aims of this study.*

- Since we include hydrogen isotope value measurements in this study, we believe that it is important to include some background information on what influences those values. Primarily the physiological drivers of fractionation by different plant species, which we observe in this study with the above and below treeline vegetation.

*L.83-86 Could you, please, summarize some of the key findings of the referenced publications, that are also relevant for this study?*

- We have changed the following text to further clarify the results from previous integrations studies (L90-97):
- "Previous studies on the integration of organic biomarkers have produced mix results, with some demonstrating spatial integration of catchment signals (Alewell et al., 2016; Feakins et al., 2018; Hemingway et al., 2016), whereas others did not observe this (Häggi et al., 2016; Ponton et al., 2014). However, these previous studies typically focused on very large river systems, which will undergo different transport processes than the first-order streams analyzed in this study. A number of these studies (Alewell et al., 2016; Feakins et al., 2018; Hemingway et al., 2016; Ponton et al., 2014) also observed seasonal differences in biomarker load in river sediments."

*L.105 What proxies were used to assess the relationships between the past treeline and climate? Are there knowledge gaps that remained unaddressed and that are addressed within the present study?*

- These previous studies included analysis of pollen, carbon and oxygen isotope values of herbivore tooth enamel, and biomarkers. This study is designed to address some of the interpretive gaps in Malinsky-Buller et al, 2021, in which we analyzed biomarkers in fluvial sediments deposited between 60-45 ka in this catchment.

*L.108 Regarding the potential of sediments at Kalavan to reconstruct the treeline-climate relationship, I assume it refers to a biomarker-based reconstruction, because it is not clear. But then, why would biomarkers be the preferred proxy instead of more established proxies, like for example plant macro-remains and pollen? Justification needs to be a bit stronger here.*

- We will make it more clear that we are referring specifically to biomarker integration processes. Plant macro-remains and pollen also are likely to be subject to uncertainty in their transportation processes, which remain unstudied in this environment, and therefore we believe biomarkers will provide a good proxy for treeline relationship in the Pleistocene sediments present in this region.

Methods:

*I recommend the authors to begin with a subsection which describes the study area in terms of climate, geology, soil types and dominant vegetation species for the two main vegetation belts and the treeline ecotone. I also recommend the authors to create another section, that could be placed last, that collates the description of the statistical methods used (significance tests and mixture models).*

- We have added references to the following publications to clarify the types of dominant vegetation in these belts.
- "Forest vegetation is predominantly oak (*Quercus macranthera*), beech (*Fagus orientalis*) and hornbeam (*Carpinus orientalis*), while above treeline alpine meadow is comprised of *Hercleum sp.* and *Senecio sp.* (Joannin et al., 2022; Volodicheva, 2002)."
    - Volodicheva, Natalya. "The Caucasus." The physical geography of northern Eurasia (2002): 350-376.
    - Joannin, Sébastien, A. Capit, V. Ollivier, O. Bellier, B. Brossier, B. Mourier, P. Tozalakian et al. "First pollen record from the Late Holocene forest environment in the Lesser Caucasus." Review of Palaeobotany and Palynology 304 (2022): 104713.

*L.119-120 Please provide a reference for the Soxhlet procedure used for lipid extraction. What intrigues me is the relatively high proportion of methanol in the solvent mixture and the long extraction time.*

*L.121 Please specify what solvent or solvent mixture was used for n-alkane separation.*

*L.121-125 Please add details on: oven temperature, use of blanks to test for lab contamination, standards used for n-alkane quantification, method used for integration of peak areas etc.*

- We have expanded the description of our extraction methods (L132-140), and added references to previous publications with the protocol used:
- "Samples were extracted using a Soxhlet apparatus with 2:1 dichloromethane:methanol for 48 hours. Following lipid extraction, n-alkanes were separated from total liquid extract by passing samples through a column of activated silica gel (1.25 g) in baked Pasteur pipettes with 2 mL hexane (non-polar fraction), 4 mL dichloromethane (slightly polar fraction) and 4 mL methanol (polar fraction). n-alkanes were quantified through the analysis of the hexane fraction. We quantified n-alkanes using a BP-5 column (30 m × 0.25 mm i.d., 0.25 μm film thickness) with He as the carrier (1.5 ml/min). Oven temperature was set at 50 °C for 1 min, ramped to 180 °C at 12 °C/min, then ramped to 320 °C at 6 °C/min and held for 4 min. (Brittingham et al., 2017; Smolen and Hren, 2023). ramped to 320 °C at 6 °C/min and held for 4 min. (Brittingham et al., 2017; Smolen and Hren, 2023). We measured a standard mixture of *n*-alkanes ($C_{20}$-$C_{33}$)

of known concentration to correct for mass dependent response decreases in longer chain *n*-alkanes."

- o Smolen, J. D., & Hren, M. T. (2023). Differential effects of clay mineralogy on thermal maturation of sedimentary n-alkanes. Chemical Geology, 634, 121572.
- o Brittingham, Alex, Michael T. Hren, Gideon Hartman, Keith N. Wilkinson, Carolina Mallol, Boris Gasparyan, and Daniel S. Adler. "Geochemical evidence for the control of fire by Middle Palaeolithic hominins." Scientific Reports 9, no. 1 (2019): 15368.

*L.125 'REF' shows a missing reference?*

- We corrected this, and added the following citation:
    - o Bush, Rosemary T., and Francesca A. McInerney. "Leaf wax n-alkane distributions in and across modern plants: Implications for paleoecology and chemotaxonomy." Geochimica et Cosmochimica Acta 117 (2013): 161-179.

Results:

*L.143-145 It would be great to see some of the most illustrative chromatograms added to the supplementary file. This would help the reader understand better the n-alkane distribution in different sets of samples.*

- We have added a supplemental file which includes 4 chromatograms, one for each type of sample analyzed (Above treeline soil, below treeline soil, above treeline stream, and below treeline stream)

*L.174 Please, add the design of the significance test to the methods section.*

- We have added references to the stastistical tests used with each mention of the results (L173, 176, 193, 195, 199, 201)

*L.152 word missing: 'between the average values of the "n-alkane?" above treeline and below…'*

- We have made this change

*I don't see any description of results obtained for the mixing model.*

- We have included a table with the constants, and a supplementary file including all of the results obtained from the mixing model

*Discussion:*

*L.184-186 Please, reference the relevant figures here.*

- We have made this change as recommended.

*L.194-208 Consider moving these paragraphs in a separate section of the methods, where you could also include information about the statistical tests used. But overall, I very much like the idea of using a mixing model to compare expected and obtained biomarker compound values.*

- We have expanded this section to further describe the results from the mixing model. We have included a new figure showing the area-based model (Fig. 6) as well as a table (Table 1) to provide the constants used. This section now reads as follows:
- L220-233 "The parameters we used for our mixing model are: 1. Satellite images (Google Earth) to map the areas covered by alpine meadow and forest vegetation throughout the Dany River catchment. 2. An estimate of net primary productivity of organic material production in grasses and trees (grams per area) (Brun et al., 2022). 3. Estimates of n-alkane production in grasses and trees in the Greater and Lesser Caucasus Mountains (grams of n-alkane per gram of organic material) (Bliedtner et al., 2018; Trigui et al., 2019). 4. End member values of δD, δ13C and ACL derived from the average hillslope soils above and below the treeline. At each sample point within the catchment, we first calculated the upstream area covered by the two dominant vegetation types within the catchment (deciduous forest and alpine meadow) (Figure 6). This area was then multiplied by the previously mentioned constants (Table 1). By multiplying these terms (area x organic mass production x n-alkane production x end member soils value), we created an n-alkane production map for the Dany River catchment. Using this method, we calculated, the amount of grass and tree n-alkanes produced on the hillslopes above the sampling locations and the expected δD, δ13C and ACL values for each stream sampling location (Figure 7a, 7c, 7e)."

*L.200 What does the phrase 'tree and grass sediment' refer to?*

- This sentence now reads ". Satellite images (Google Earth) to map the areas covered by alpine meadow and forest vegetation throughout the Dany River catchment."

*L.210-212 Please reference the relevant figure for this statement.*

- We have added references to Figure 7 here (L233, L238)

*L.212 I assume 'deciduous-sourced n-alkanes' refers to deciduous trees, but it's a bit ambiguous, because there are also deciduous herbaceous plants. Could you also include what the dominant deciduous species in the forest are?*

- In order to make this clearer to the reader, we refer to these as "n-alkanes sourced from below-treeline vegetation". We have added a section in the background describing the vegetation structure and dominant vegetation types in the forest (Oak/Beech/Hornbeam) .
- L234-237: "We compared the results of our mixing model with the measured $\delta D$, $\delta^{13}C$ and ACL in the streams. Stream sediment samples collected above the treeline (from ~2000-2600 masl) fall within the range of expected values, however, samples below the treeline consistently over-sample $n$-alkanes sourced from below treeline vegetation"

*L.228 Please include the distance from the study site of the lacustrine core that was analyzed for pollen.*

- L254: "However, analysis of pollen from a lake core nearby (~ 5km from the Dany catchment) in the Areguni Mountains shows a gradual shift over the last 4000 years from a grass-dominated landscape to the deciduous forest present today"

*L 233-234 This general statement needs a reference.*

- We have altered the text to make it clearer that we are specifically referring to the results of this study
- L259: "Since $n$-alkanes in the first order stream in this study do not quantitively integrate $n$-alkanes based on the upstream area of different vegetation types, this likely precludes the use of $n$-alkanes as a tool to reconstruct vertical treeline movement in this setting"

*L.261-263 As an additional research direction, perhaps collecting water samples for lipid analysis could help clarifying the role of transport and depositional processes.*

- We appreciate the suggestion from the reviewer, this would be an interesting direction for future research

*Figures: most of them are blurry and should be uploaded in a better resolution.*

- We have re-uploaded the figures in higher resolution

*Fig. 1 Could you, please, specify the source of the satellite images? Also, I would find it more relevant if the figure included a close-up of the studied catchment with sampling points superposed on vegetation types.*

- We have included a reference to the DEM used (ASTER). A new figure has been added (Figure 7) which includes a close-up of the catchment with sampling points

*Figures 3-5. I assume the green and red rectangles are soil samples (although it is not clear, and also not colorblind-friendly). But it should also be clarified which of the stream samples (blue triangles) were taken from above and from below the treeline respectively.*

- In order to de-clutter this figure, we have removed the stream sediment samples from these three plots, and changed the color palette. These samples are plotted along their elevation gradient in Figure 7, this should make the data visually clearer.

*Figure 6. Please make it larger, and also increase the resolution, because the labels are hardly visible.*

- We have included a higher resolution version of this figure (now figure 7)